# Nitrogen Containing Heterocycles as Anticancer Agents: A Medicinal Chemistry Perspective

**DOI:** 10.3390/ph16020299

**Published:** 2023-02-14

**Authors:** Adarsh Kumar, Ankit Kumar Singh, Harshwardhan Singh, Veena Vijayan, Deepak Kumar, Jashwanth Naik, Suresh Thareja, Jagat Pal Yadav, Prateek Pathak, Maria Grishina, Amita Verma, Habibullah Khalilullah, Mariusz Jaremko, Abdul-Hamid Emwas, Pradeep Kumar

**Affiliations:** 1Department of Pharmaceutical Sciences and Natural Products, Central University of Punjab, Ghudda, Bathinda 151401, India; 2Pharmacology Research Laboratory, Faculty of Pharmaceutical Sciences, Rama University, Kanpur 209217, India; 3Laboratory of Computational Modeling of Drugs, Higher Medical and Biological School, South Ural State University, 454008 Chelyabinsk, Russia; 4Bioorganic and Medicinal Chemistry Research Laboratory, Department of Pharmaceutical Sciences, Sam Higginbottom University of Agriculture, Technology and Sciences, Prayagraj 211007, India; 5Department of Pharmaceutical Chemistry and Pharmacognosy, Unaizah College of Pharmacy, Qassim University, Unayzah 51911, Saudi Arabia; 6Smart-Health Initiative and Red Sea Research Center, Division of Biological and Environmental Sciences and Engineering, King Abdullah University of Science and Technology, Thuwal 23955-6900, Saudi Arabia; 7Core Labs, King Abdullah University of Science and Technology, Thuwal 23955-6900, Saudi Arabia

**Keywords:** heterocyclic, anti-cancer, FDA, nitrogen-containing heterocyclic, biological activity

## Abstract

Cancer is one of the major healthcare challenges across the globe. Several anticancer drugs are available on the market but they either lack specificity or have poor safety, severe side effects, and suffer from resistance. So, there is a dire need to develop safer and target-specific anticancer drugs. More than 85% of all physiologically active pharmaceuticals are heterocycles or contain at least one heteroatom. Nitrogen heterocycles constituting the most common heterocyclic framework. In this study, we have compiled the FDA approved heterocyclic drugs with nitrogen atoms and their pharmacological properties. Moreover, we have reported nitrogen containing heterocycles, including pyrimidine, quinolone, carbazole, pyridine, imidazole, benzimidazole, triazole, β-lactam, indole, pyrazole, quinazoline, quinoxaline, isatin, pyrrolo-benzodiazepines, and pyrido[2,3-d]pyrimidines, which are used in the treatment of different types of cancer, concurrently covering the biochemical mechanisms of action and cellular targets.

## 1. Introduction

Carcinoma is the abnormal growth of normal cells that typically grow beyond their original boundaries, invade surrounding areas, spread to other organs, and result in metastasis, which is one of the main causes of cancer-related death, the second most common cause of deaths across the globe [1]. Around 10.0 million cancer-related fatalities (9.9 million excluding squamous cell carcinoma) and 19.3 million new cases of cancer (18.1 million excluding squamous cells carcinoma) were estimated globally by 2020. Up to 25% of cancer cases are caused by cancer-causing illnesses such as hepatitis as well as human papillomavirus infections. The most common malignancies in both genders are breast, lung, stomach, colorectal, thyroid, liver, and ovarian. The most fatal cancers are lung (1.8 million), liver (830,000), stomach (769,000), breast cancer (627,000), and colorectal (935,000). The most commonly diagnosed cancers worldwide are lung (2.2 million), breast (2.09 million), colorectal (1.9 million), prostate (1.28 million), skin (1.04 million), and stomach (1.04 million). The prevalence of cancer is rising worldwide, burdening people and families emotionally and financially [2,3,4,5].

In wealthy nations, cancer has become one of the leading causes of mortality. A DNA mutation can cause cancer to develop. The etiology of the gene can be either adopted or hereditary. Both genetic and epigenetic variables are involved. The co-carcinogenic responses, hormonal impacts, and many other epigenetic variables are tumor promoters. Conversion of protooncogene to oncogene and the inactivation of tumor-suppressing genes are the two genetic factors responsible for cancer. Cancer cells can divide quickly or slowly, as in the cases of plasma tumor cells and Burkitt’s lymphoma, respectively. In the cancer cell growth, proliferation is the crucial element, which is very quick compared to the regular cells. When cancerous cells proliferate out of control, alterations occur in the proteins and growth factors, telomerase expression, tumor-associated angiogenesis, and intracellular signaling pathways that regulate apoptosis and cell cycles [6]. Sulfur mustards were utilized in battle throughout the First World War, producing marrow aplasia during that period and later being applied to the chemoprevention. In the last 30–50 years, several other types were developed, including folate analogues, pyrimidine inhibitors, and purine inhibitors. Depending on the type of cancer, different anti-cancer medicines were chosen for the treatment. Surgery, which is similar to plucking a seed from its shell, can eradicate benign tumors, typically referred as nonmalignant [7,8].

Heterocyclic are substances where the ring carbon atom has been replaced by one of the other three elements, oxygen, nitrogen, or sulphur, in the parent scaffold. The existence of the modified atoms as well as the size of the scaffold in the compound affects its physical and chemical properties. Modifications to a molecule’s heterocyclic ring structure can alter its anti-inflammatory, antibacterial, anti-tumor, antiviral, and antifungal properties. In nature, nitrogen-containing heterocyclic compounds are widely distributed and serve as the basis for many different substances, including alkaloids, vitamins, hormones, dyes, antibiotics, herbicides, and pharmaceuticals [9,10,11]. There are few examples of naturally occurring molecules with nitrogen atoms, including morphine, caffeine, nicotine, thiamine, and atropine, known as alkaloids. The number of nitrogen atoms found inside the ring, such as three, four, five, as well as six, is used to categorize these molecules. Pyrrole and azoles are five membered rings containing one nitrogen, while imidazole and pyrazole have two atoms of nitrogen. Pyridine, a six-membered ring with one nitrogen atom, and pyrimidine, a six-membered ring with two nitrogen atoms, are the best examples of nitrogen containing heterocycles [12].

For many years, nitrogen-containing heterocycles have attracted attention of scientists due to their structural variety and biological importance. The present study covers the most recent developments in nitrogen-containing heterocyclic compounds as potential cancer chemotherapeutics. With approximately 60% of unique small-molecules containing a nitrogen heterocyclic, a quick glance through FDA archives demonstrates the structural significance of nitrogen-based heterocycles in drug design. Due to the formation of hydrogen bonds between these heteroatoms and DNA, complexes containing heteroatoms are more stable. In reality, the strength of the binding between DNA and heterocyclic compounds corresponds with the anti-cancer impact [13,14,15]. Natural products, pharmaceuticals, organic materials, sensitizers, copolymers, dyestuff, dyes and corrosion inhibitors all contain nitrogen heterocycles in their skeleton [16,17,18,19,20].

In addition to their important structural role in herbal drugs, e.g., codeine, morphine, vinblastine, reserpine, procaine, papaverine, emetine, and cardiac glycosides, nitrogen-containing heterocyclic compounds are also widely present in synthetic drugs, e.g., azidothymidine, chlorpromazine, antipyrine, metronidazole, diazepam, captopril, isoniazid, chloroquinine, and barbituric acid [21]. Numerous active medicines and natural compounds contain heterocyclic scaffolds as their fundamental nuclei. According to statistics, ≥85% physiologically active molecules are heterocycles or contain one nitrogen atom in their intricate structures [22]. We searched ‘‘Nitrogen containing heterocyclic compounds’’ on ChEMBL (https://www.ebi.ac.uk/chembl/, accessed on 22 November 2022), an open access biological database, and found 2,331,700 compounds on 467 targets. With the help of ChEMBL, we plotted a graph between the molecular weight of nitrogen containing heterocyclic compounds with log P values and color, which indicated the violation of rule of 5 (RO5). LogP values of the reported molecules were directly correlated with their molecular weight (Figure 1). As there are so many molecules violating RO5, to explore the correlation of RO5 with activity, we have plotted a heatmap of 929 nitrogen containing molecules violating RO5 against different biological targets, but certain molecules violating RO5 have biological activities (Figure 2). Drug development commonly employs Lipinski’s rule of five. This criterion makes it possible to determine whether a bioactive molecule would likely possess the physical and chemical traits necessary for oral bioavailability. According to the Lipinski rule, certain physicochemical features determine how a medicine will behave in terms of absorption, distribution, metabolism, and excretion. The PChEMBL average value indicates the activity count for particular compounds [23].

## 2. Nitrogen-Containing FDA-Approved Anti-Cancer Drugs

There are many FDA approved drugs available on the market from natural as well as synthetic sources. Some of them with nitrogen atom are covered in Table 1 with their cellular target and year of approval.

**Table 1 pharmaceuticals-16-00299-t001:** Nitrogen-containing FDA-approved drugs.

S. No.	Name	Structure	Mode of Action	Drug Target	Approval Year	References
1.	Alectinib Hydrochloride	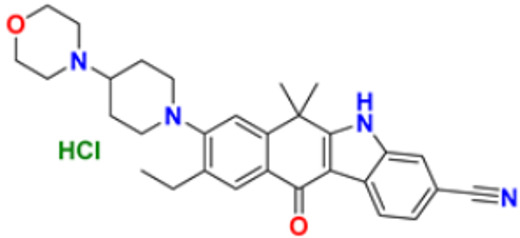	It prevents the multiplication of cancer cells by blocking the action of abnormal proteins.	Anaplastic lymphoma kinase (ALK) Inhibitor	2015	[24]
2.	Brigatinib	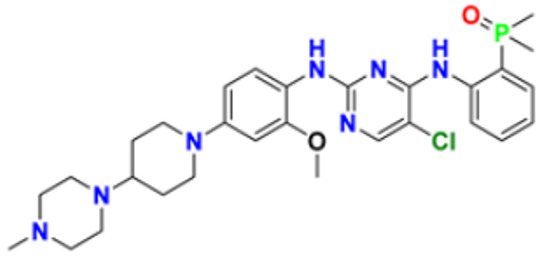	It inhibits the ability of the phosphate group to bind ALK and inhibits the activity of proteins STAT3, A.K.T., ERK1/2, and S6 both in vitro and in vivo.	Anaplastic lymphoma kinase (ALK) Inhibitor	2017	[25]
3.	Lorlatinib	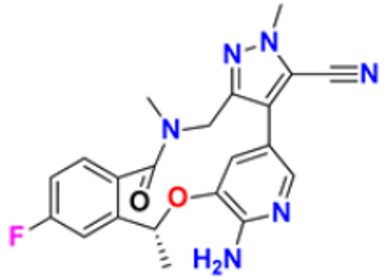	Inhibits will disrupt the ALK and ROS1- mediated signaling, further inhibiting the growing tumor cells in ALK and ROS1 cells.	Anaplastic lymphoma kinase (ALK) Inhibitor	2018	[26]
4.	Entrectinib	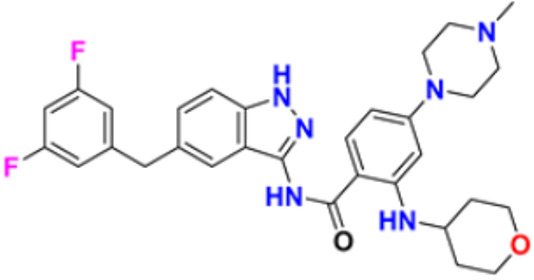	It interferes with the growth of cancer cells, which are eventually destroyed.	Anaplastic lymphoma kinase (ALK) Inhibitor	2019	[27]
5.	Midostaurin	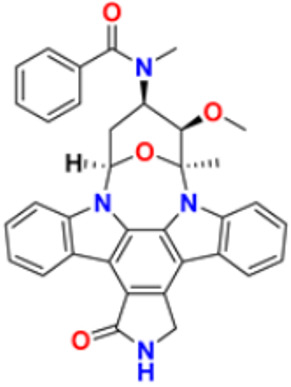	It Prevents the multiplication of cancer cells by blocking the action of abnormal proteins.	Fms-like tyrosine kinase (FLT3) inhibitor	2017	[28,29]
6.	Gilteritinib fumarate	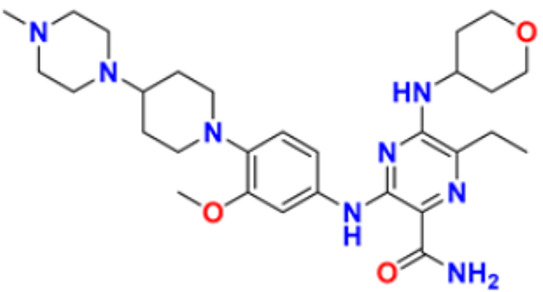	Blocks the action of naturally occurring substances that promote cancer cell growth.	Fms-like tyrosine kinase (FLT3) inhibitor	2018	[30]
7.	Quizartinib	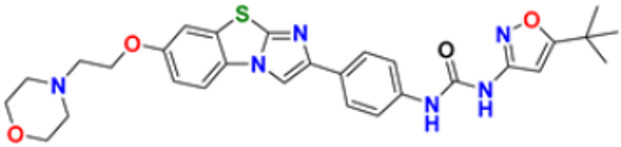	It inhibits cancer cell proliferation which leads to the death of the cells.	Fms-like tyrosine kinase (FLT3) inhibitor	2018	[31]
8.	Pexidarinib	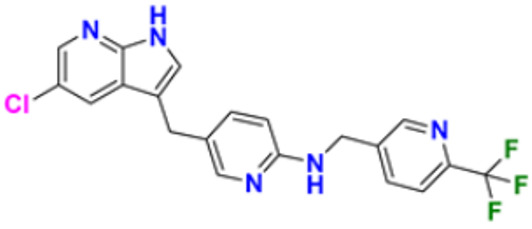	Inhibits the colony-stimulating factor (CSF1)/CSF1 receptor pathway	Fms-like tyrosine kinase (FLT3) inhibitor	2019	[32]
9.	Osimertinib mesylate	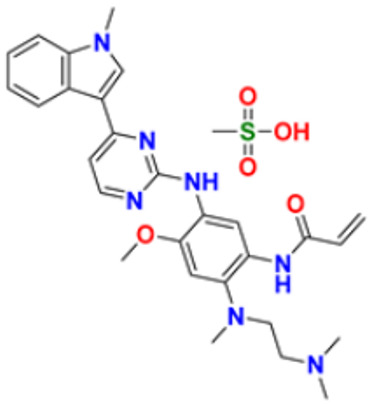	The abnormal protein’s ability to cause cancer cells to grow is blocked, which may also help tumors get smaller and slow down the spread of cancer cells.	Epidermal growth factor (EGF) receptor inhibitors	2018	[33]
10.	Olmutinib	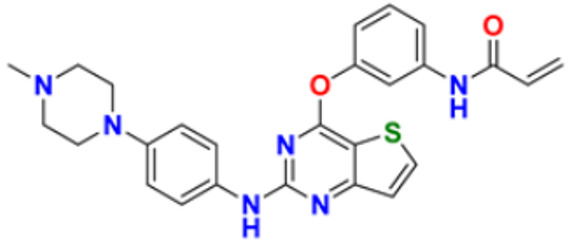	Restricting the mutant form of E.G.F.R. causes the death of E.G.F.R. expressing tumor cells.	Epidermal growth factor (EGF) receptor inhibitors	2015	[34]
11.	Neratinib maleate	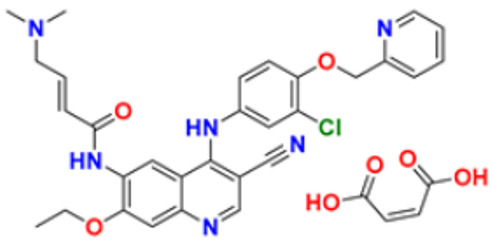	The autophosphorylation is prevented on tyrosine residues receptor, and the Oncogenic signaling was reduced through mitogen-activated protein kinase and Akt pathways.	Epidermal growth factor (EGF) receptor inhibitors	2017	[35]
12.	Dacomitinib	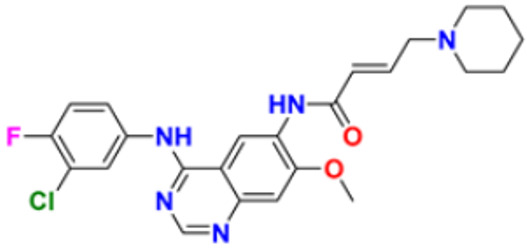	The E.G.F.R. activating mutations like exon 19 deletion or the exon 21 L858R substitution mutation and E.G.F.R. proteins like EGFR/HER1, HER2, and HER4 activities are blocked.	Epidermal growth factor (EGF) receptor inhibitors	2018	[36]
13.	Pyrotinib maleate	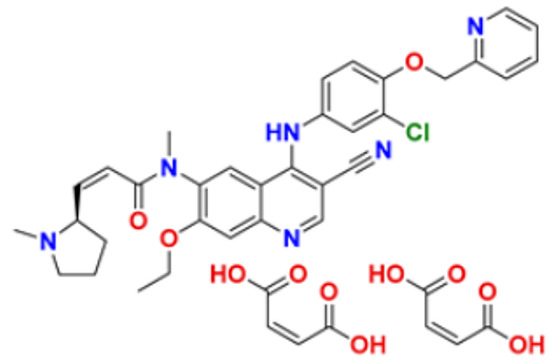	Covalently binds to the intracellular kinase domain of HER1, HER2, and HER4 to block downstream signaling pathways, inhibits autophosphorylation, and prevents the creation of homologous/heterodimers of the HER family.	Epidermal growth factor (EGF) receptor inhibitors	2018	[37]
14.	Almonertinib mesylate	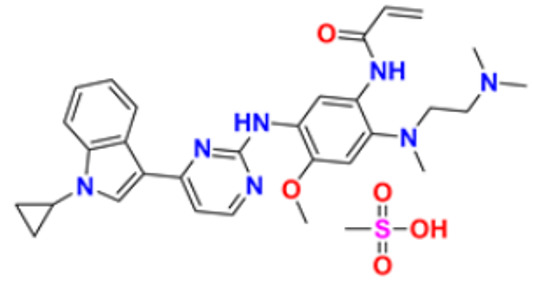	It inhibits E.G.F.R. tyrosine kinase targeting EGFR-sensitizing and T790M resistance mutations.	Epidermal growth factor (EGF) receptor inhibitors	2020	[38]
15.	Lenvatinib mesylate	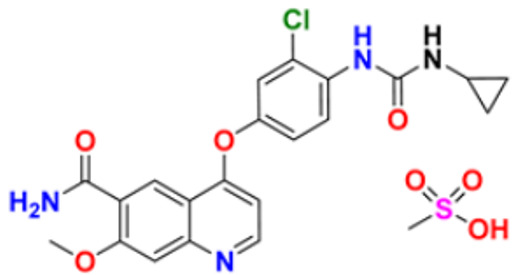	The kinase activities of vascular endothelial growth factor (VEGF) receptors VEGFR1 (FLT1), VEGFR2 (K.D.R.), and VEGFR3 (FLT4) are inhibited.	Vascular endothelial growth factor (VEGF) inhibitors	2015	[39]
16.	Tivozanib	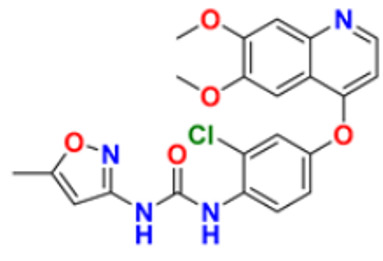	The phosphorylation of V.E.G.F.R. (vascular endothelial growth factor) receptors (V.E.G.F.R.)-1, VEGFR-2, and VEGFR-3, as well as kinases including c-kit and platelet-derived growth factor beta (P.D.G.F.R.), are suppressed.	Vascular endothelial growth factor (VEGF) inhibitors	2021	[40]
17.	Fruquintinib	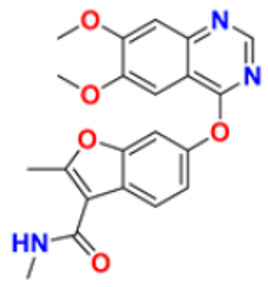	It prevents the VEGF from causing the phosphorylation of V.E.G.F.R.s 1, 2, and 3, which may reduce the proliferation, migration, and survival of endothelial cells, the development of microvessels, and the proliferation and death of tumor cells.	Vascular endothelial growth factor (VEGF) inhibitors	2020	[41]
18.	Anlotinib dihydrochloride	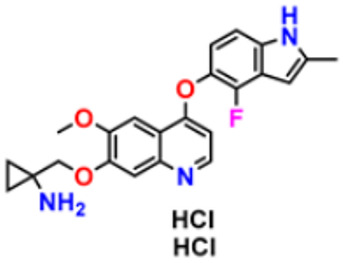	It inhibits the development of tumor cells by inhibiting the pathways like PI3K/AKT, RAS/MAPK, and PLCy/PKC by using the receptors: V.E.G.F.R., F.G.F.R., and P.D.G.F.R.	Vascular endothelial growth factor (VEGF) inhibitors	2020	[42]
19.	Acalabrutinib	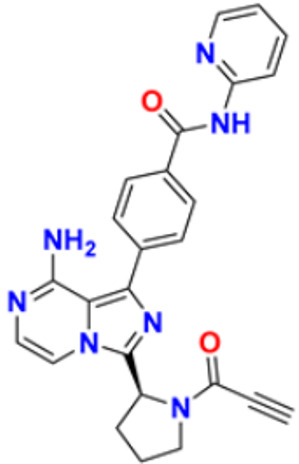	Bruton Tyrosine Kinase inhibitor that stops B cells from chemotactic, proliferating, moving, and adhesion of B cells	Tyrosine kinase inhibitor(tki)	2017	[43]
20.	Zanubrutinib	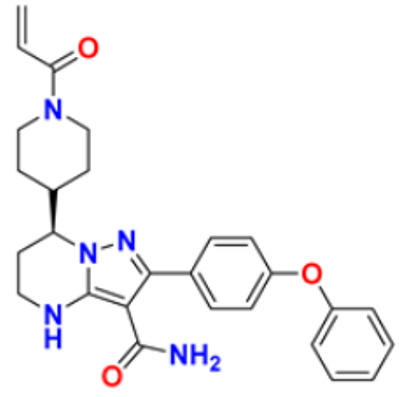	Inhibits the growth and survival of cancerous B cells to shrink the size of the tumor in mantle cell lymphoma	Tyrosine kinase inhibitor(tki)	2021	[44]
21.	Fedratinib	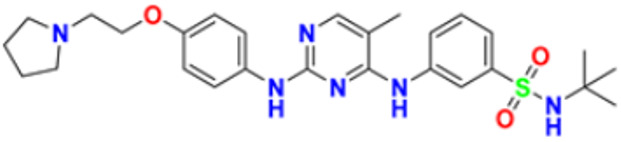	Inhibits cell division and induces apoptosis	Janus kinase 2 (JAK2) inhibitors	2019	[45]
22.	Copanlisib	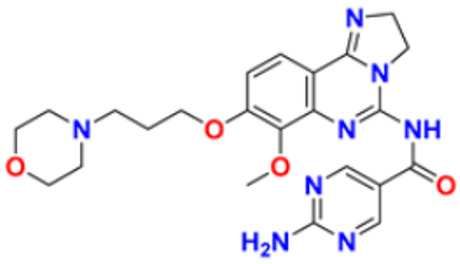	Apoptosis-induced tumor cell death and primary malignant B cell line growth suppression.	Phosphatidylinositol 3-hydroxy kinase inhibitors	2014	[46,47]
23.	Mocetinostat	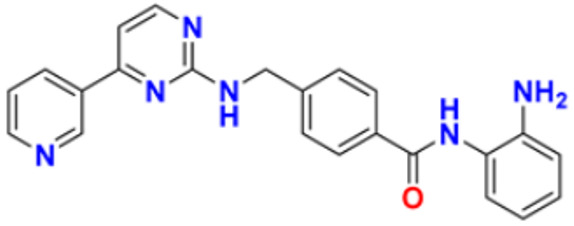	Acts by turning on tumor suppressor genes that have been inappropriately turned off	Hydroxamic acid-based hdac Inhibitors (hdaci)	2014	[48,49]
24.	Duvelisib	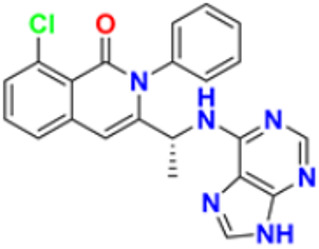	Inhibit isoform gamma, essential for cytokine signaling and the pro-inflammatory response, and inhibit the isoform delta of PI3K(*phosphoinositide3-kinase)*, which is required for cell proliferation and survival.	Phosphatidylinositol 3-hydroxy kinase inhibitors	2018	[50]
25.	Oxaliplatin	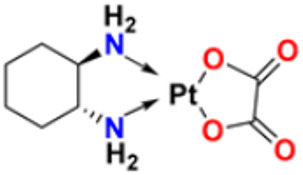	Inhibits D.N.A. synthesis and transcription by binding preferentially to the guanine and cytosine moieties of D.N.A.	Platinum derivative	2004	[51,52]
26.	Irinotecan	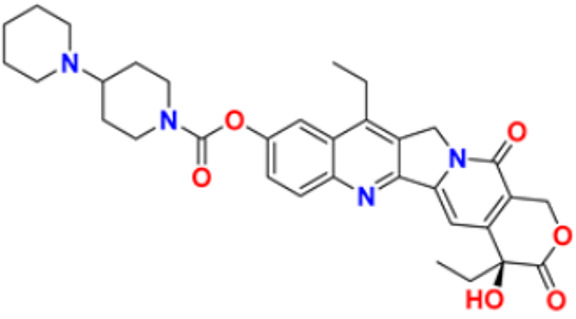	Binds to the topoisomerase I-DNA complex and prevents the D.N.A. strand from relegating, which then forms a ternary complex disrupting the replication fork and moving slowly, leading to breakage of lethal double-stranded DNA.	Topoisomerase I inhibitor	2000	[53,54]
27.	Cytarabine	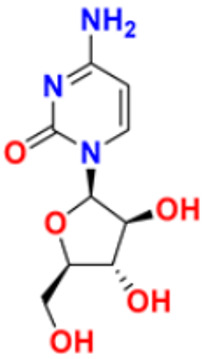	It prevents cancer cells from generating and repairing the D.N.A., which they require to thrive and proliferate.	Antineoplastic anti-metabolite	1999	[55,56]
28.	Gemcitabine	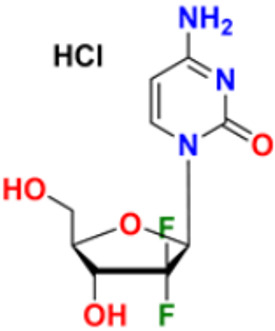	Inhibits D.N.A. synthesis	Pyrimidine nucleoside antimetabolite	2011	[57,58]
29.	Trabectedin	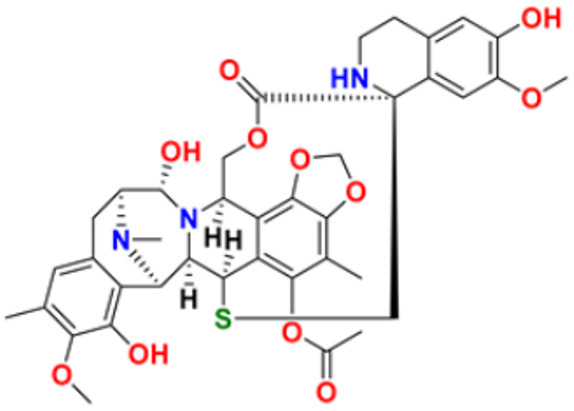	Inhibition of trans-activated transcription and the interaction with D.N.A. repair proteins	Alkylating agent	2015	[59,60]
30.	Kahalalide F	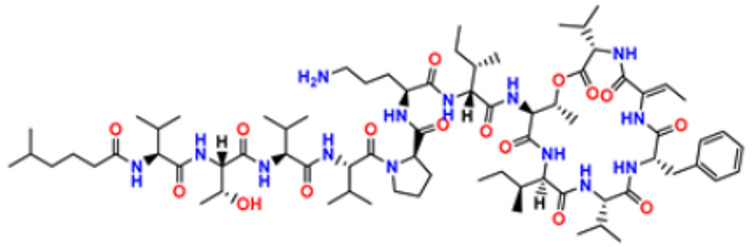	Induces oncosis	Lysosomes	2004	[61,62]
31.	Salinosporamide A	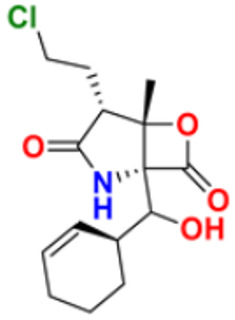	Covalently modifies the threonine residues in the 20S proteasome’s active site to decrease proteasome activity.	Inhibitor of the 20S proteasomes	2004	[63]
32.	Osimertinib	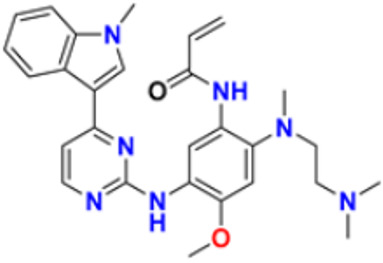	Binds to specific E.G.F.R. mutants (T790M, L858R, and exon 19 deletions), which are more prevalent in non-small cell lung cancer (N.S.C.L.C.) following the first-line EGFR-TKI treatment.	Epidermal growth factor receptor (E.G.F.R.) tyrosine kinase inhibitor (T.K.I.)	2015	[64]
33.	TZT1027 (Soblidotin)	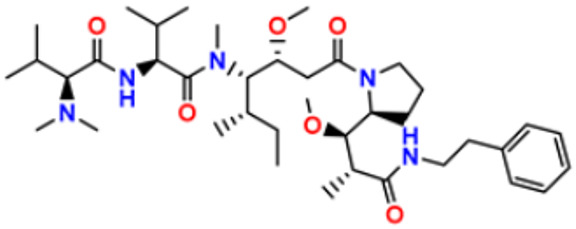	Inhibits tubulin polymerization	Inhibit microtubule polymerization	2011	[65]
34.	Selinexor	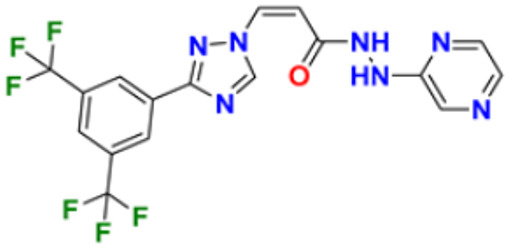	XPO1 is selectively inhibited, and tumor-suppressing proteins are activated, causing tumor cell death	Inhibiting exportin 1 (XPO1)	2019	[66,67]
35.	Bendamustine	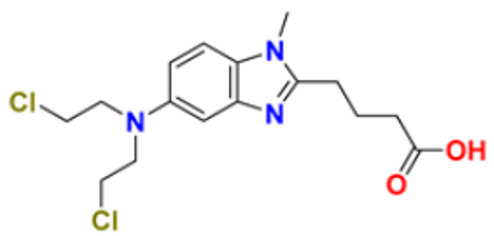	Eradicates current cancer cells and restricts the development of new cancer cells.	Alkylation	2008	[68]
36.	Gefitinib	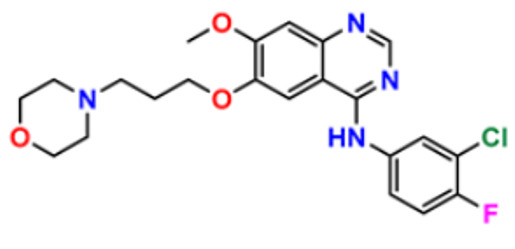	It prevents a variety of tyrosine kinases linked to transmembrane cell surface receptors from becoming phosphorylated intracellularly, including those linked to the epidermal growth factor receptor (EGFR-TK).	Epidermal growth factor receptor tyrosine kinase inhibitors (EGFR-tkis)	2003	[69]
37.	Abemaciclib	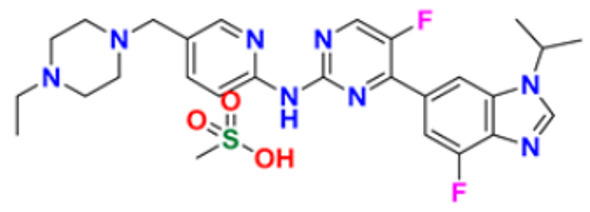	The action of an abnormal protein is blocked, which causes the cancer cell’s signals to multiply	Blocks the activity of CDK4 and CDK6 proteins (CDK4/6)	2017	[70,71]
38.	Ribociclib	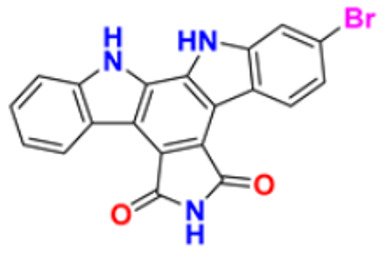	Cell-cycle progression is blocked by acting directly on the cyclin D–CDK4/6–p16–Rb pathway	Cyclin-dependent kinase 4 and cyclin-dependent 6 (CDK 4 and CDK 6) Inhibitor	2018	[72]
39.	Mitomycin C	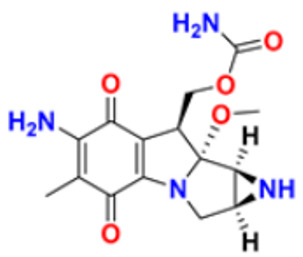	Causes cross-linking and inhibition of D.N.A. synthesis and function	Double-stranded D.N.A. alkylating agent	1974	[73]
40.	Porfimer	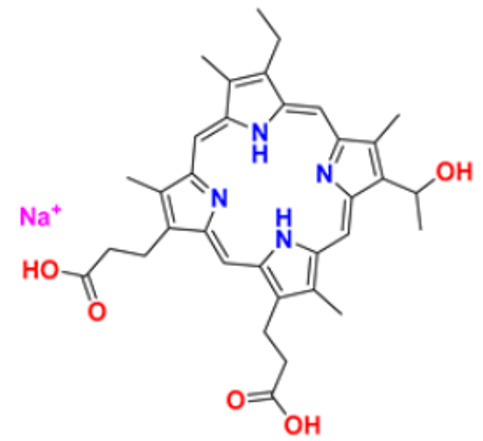	When activated by laser light, a cytotoxic activity that depends on oxygen and light causes the release of thromboxane A2, which in turn causes vascular necrosis and oxygen-free radicals.	Photodynamic therapy agent High-affinity immunoglobulin gamma Fc receptor I antagonist	2003	[74]
41.	Topotecan	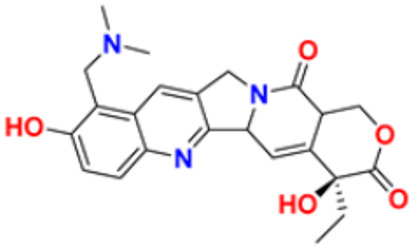	D.N.A. damage occurs by prevention of topoisomerase from relegating cleaved D.N.A. strand	Novel topoisomerase-1 inhibitor	1996	[75]
42.	Idelalisib	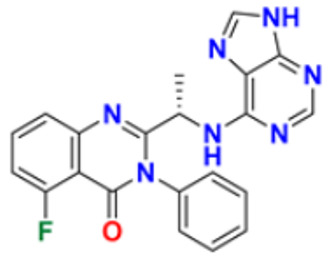	Prevents proliferation and induces apoptosis in cell lines derived from malignant B-cells and in primary tumor cells	PI3K inhibitor	2014	[76,77]
43.	Acalabrutinib	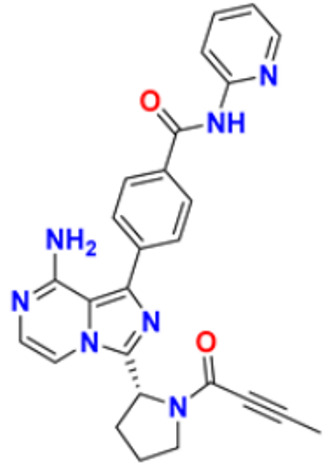	It inhibits B.C.R. signaling by reduced phosphorylation of PLCγ2.	Bruton tyrosine kinase (btk) inhibitor	2017	[78]
44.	Cabozantinib	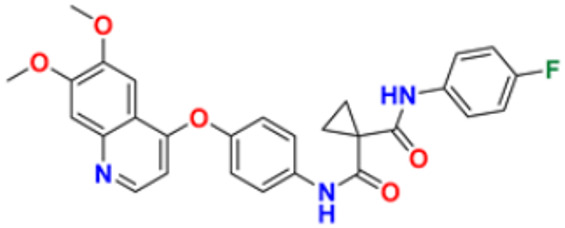	Resistance of V.E.G.F.R. inhibitor via the c-MET axis is decreased by inhibiting V.E.G.F.R. and c-MET	Tyrosine kinase inhibitor	2016	[79,80]
45.	Lenvatinib	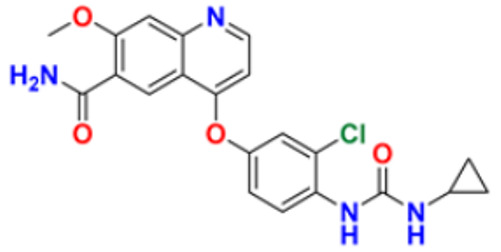	The kinase activities of vascular endothelial growth factor (VEGF) receptors VEGFR1 (FLT1), VEGFR2 (K.D.R.), and VEGFR3 (FLT4) are inhibited.	Multi-receptor Tyrosine Kinase Inhibitor.	2018	[81,82]
46.	Palbociclib	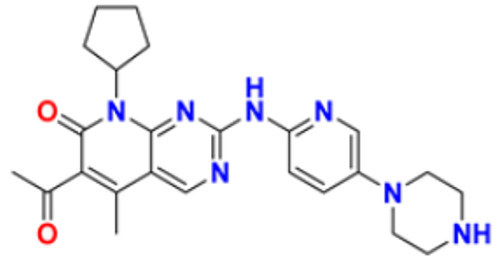	Blocking cell cycle progression from G1 into S phase and inhibits the phosphorylation of retinoblastoma (Rb) protein,	Proteins cyclin-dependent kinase 4 and 6 (CDK4 and CDK6) Inhibitor	2016	[83,84]
47.	Trametinib	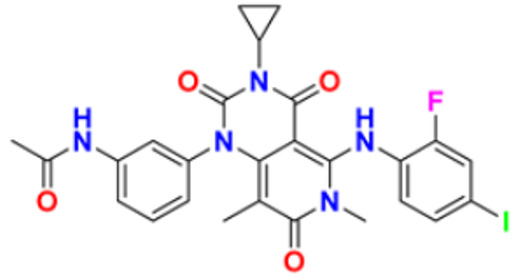	Decreases cell growth, causes G1 cell-cycle arrest and causes apoptosis.	M.E.K. 1 and M.E.K. 2 protein (kinase) Inhibitor	2013	[85,86,87]
48.	Afatinib	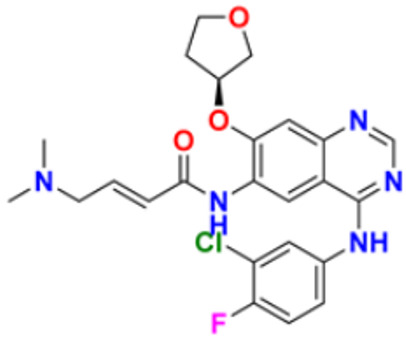	Suppresses mTORC1, which initiates apoptosis of the cancer cells.	Tyrosine kinase inhibitor	2016	[88,89]
49.	Dabrafenib	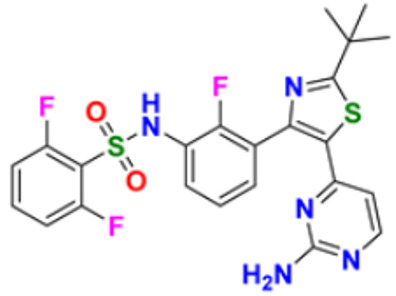	The major inhibitors of the B.R.A.F., R.A.F. proteins and C.R.A.F., are through the competitive binding of ATP and the active conformation of B.R.A.F. kinase.	Mutated B.R.A.F. proteins (kinase) Inhibitor	2013	[90,91]
50.	Gilteritinib	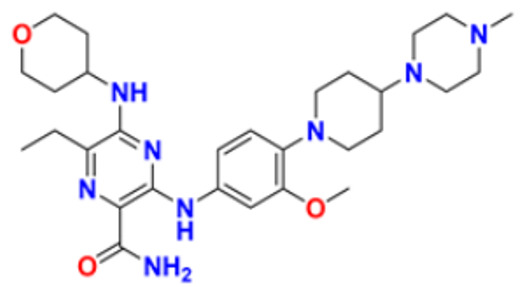	Inhibits FLT3 receptor signaling and proliferation	Tyrosine kinase inhibitor	2018	[92,93]
51.	Alpelisib	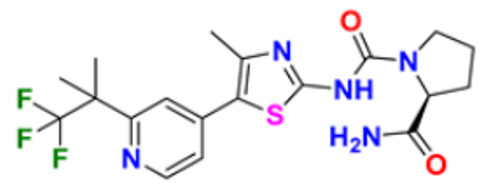	Blocks the catalytic subunit of PI3K, class I PI3K p110, a lipid kinase involved in several biological processes, including proliferation, survival, differentiation, and metabolism	Phosphatidylinositol 3-kinase (PI3K) inhibitor	2019	[94]
52.	Binimetinib	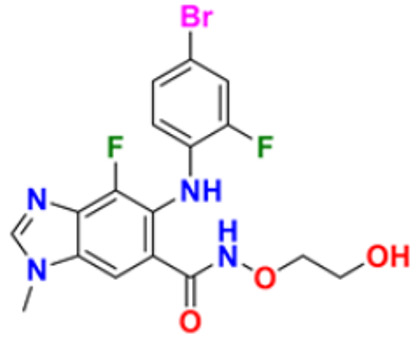	The activation of MEK1/2-dependent effector proteins and transcription factors is prevented by inhibiting MEK1/2	Mitogen-activated protein kinase kinase 1, Dual specificity mitogen-activated protein kinase kinase 2 Inhibitor	2018	[95,96]
53.	Crizotinib	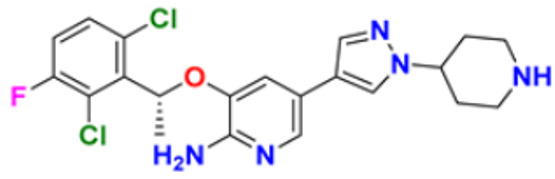	Blocks certain chemical messengers that tell cells to grow.	Small-molecule kinase inhibitor ALK, c-MET, and ROS1	2022	[97,98]
54.	Dactinomycin	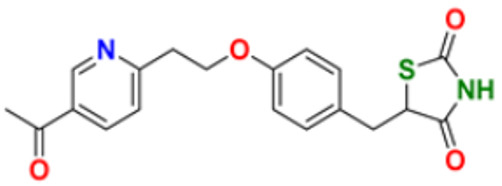	D.N.A. intercalation and the stabilization of DNA-topoisomerase I and II cleavable complexes	Potent transcription inhibitor	2009	[99]
55.	Erlotinib	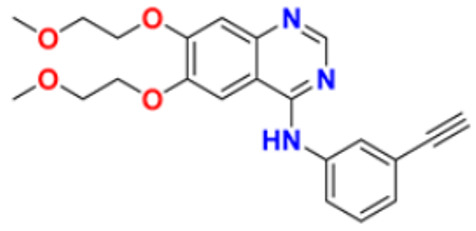	The intracellular phosphorylation of tyrosine kinase associated with the epidermal growth factor receptor (E.G.F.R.) is inhibited	Inhibition of epidermal growth factor receptor (E.G.F.R.)	2004	[100]
56.	Ibrutinib	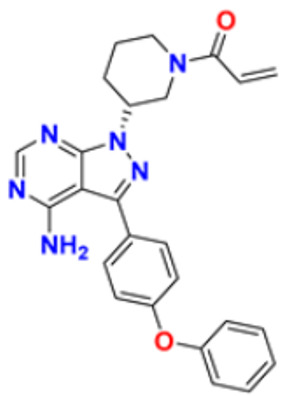	It inhibits Btk’s enzyme activity by forming a covalent bond with a cysteine residue (CYS-481) at the active site.	Bruton tyrosine kinase inhibitor	2022	[101,102]
57.	Lapatinib	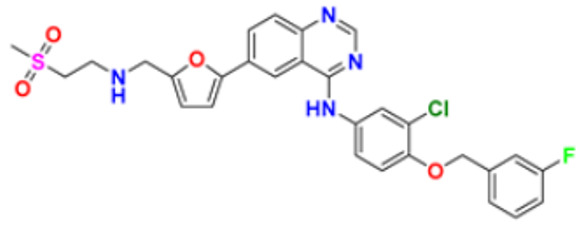	It works by competitively binding to the intracellular ATP-binding site of the receptor, inhibiting the tyrosine kinase domains of the human epidermal growth factor receptor (HER)-2 and the epidermal growth factor receptor.	HER2 and E.G.F.R. antagonist	2007	[103]
58.	Larotrectinib	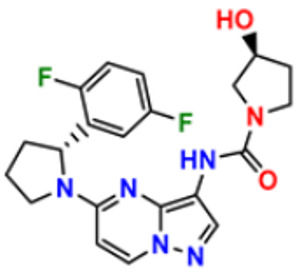	The activity of T.R.K. proteins is disrupted, which is caused by fusion in a family of genes known as N.T.R.K.	T.R.K. (Tropomyosin Receptor Kinase) inhibitor	2018	[104,105]
59.	Mercaptopurine	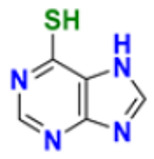	De novo purine synthesis is inhibited and acts as an antiproliferative agent by interfering with protein, R.N.A. and D.N.A. synthesis and induces apoptosis.	Hypoxanthine-guanine phospho ribosyl transferase, Amido phospho ribosyl transferase, Inosine-5′-monophosphate dehydrogenase Inhibitor	2014	[106]
60.	Methotrexate	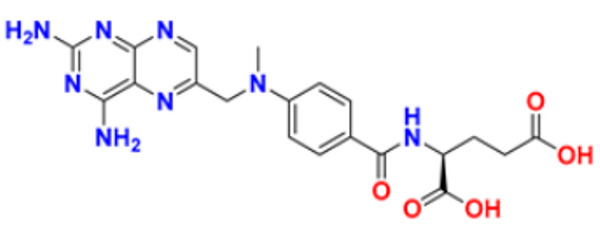	The enzymes responsible for nucleotide synthesis are aminoimidazole carboxamide ribonucleotide transformylase (A.I.C.A.R.T.), dihydrofolate reductase, thymidylate synthase, and amido phosphor ribosyl transferase are inhibited.	Thymidylate synthase and Dihydrofolate reductase Inhibitors	1999	[107,108]
61.	Nilotinib	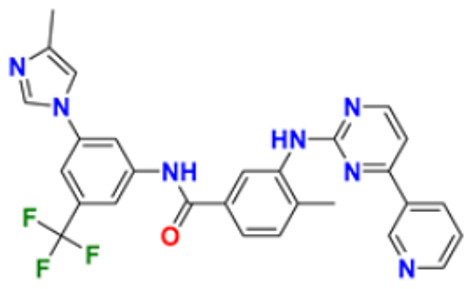	Stabilizes the Abl protein’s kinase domain’s inactive conformation by binding to it.	Tyrosine kinase inhibitor	2007	[109]
62.	Nilutamide	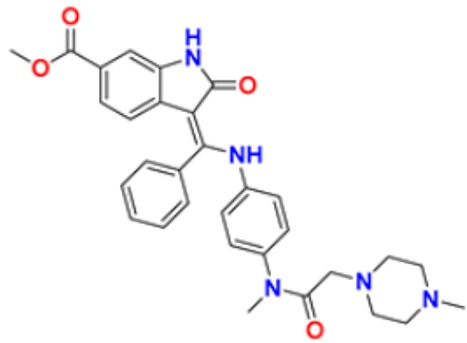	The action of androgens of testicular and adrenal origin that stimulate the growth of normal and malignant prostatic tissue is blocked	Androgen receptor antagonist	2016	[110]
63.	Olaparib	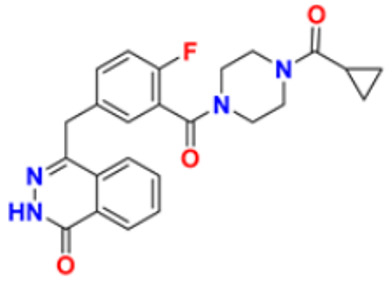	Blocks the repair of single-strand D.N.A. breaks by inhibiting poly(ADP-ribose) polymerase.	Poly (ADP-ribose) polymerase (*PARP*) inhibitor	2014	[111,112]

## 3. Current Advances in Nitrogen Containing Heterocycles as Anticancer Agents

### 3.1. Pyrimidine Derivatives as Anticancer Agents

Pyrimidines gained popularity in the history of organic chemistry as “m-Diazine” which is the product of uric acid catabolism. Brugnatelli discovered the first pyrimidine derivative, alloxan, in 1818 while oxidising uric acid with nitric acid. The heterocyclic six membered aromatic ring of pyrimidine contains nitrogen atoms in 1st and 3rd position. The melting and boiling temperatures of pyrimidine are 22.5 °C and 124 °C, respectively [113,114].

Fathalla et al. (2012) synthesized 10 pyrimidine derivatives and evaluated their antitumor activity against a liver cancer (HepG2) cell line by a comparison with the well-known anticancer drugs 5-Flurouracil and Doxorubicin. While comparing the synthesized compounds, growth inhibition effectiveness was shown on the tested tumor cell line at doses between 1 and 10 µg/mL. The most potent compound in this study was found to be compound **1** with the IC_50_ value of 3.56 µg/mL, whereas doxorubicin and 5-flurouracil were having IC_50_ values 3.56 µg/mL and 5 µg/mL, respectively [115]. Ahmed et al. (2020) synthesized and evaluated the anti-tumor activity of 16 nitrogen heterocyclic compounds bearing a pyrimidine moiety. The newly developed pyrimidine derivatives were tested for in vitro anti-proliferative activity against human liver (HepG2), breast (MCF7), and normal fibroblast (WI–38) cell lines, and their efficacy was compared to Doxorubicin. Among all the tested compounds, compound **2** showed excellent anticancer activity with IC_50_ values of 7.36, 10.76, and 6.7 µM respectively, whereas IC_50_ values of doxorubicin were 4.5, 4.1, and 6.7 µM (WI-38) respectively [116]. Gupta et al. (2022) studied the anticancer activity of spiroisoquinoline-pyrimidine derivatives against the MCF-7 cancer cell line. Out of these, compound **3** having an ethoxy group from the acetylene molecule was found to be the most potent cytotoxic agent with an IC_50_ value of 98.8 µM as compared to the reference Doxorubicin. An MTT assay was carried out at the concentration of 50 µM and it showed 60% of cell viability with control doxorubicin having 100% cell viability [117]. Al-Issa (2013) synthesized fused pyrimidines and tested them in vitro anti-tumour activity against human cancer cell line HEPG2. Out of these compounds, the most potent anticancer activity was shown by the compound **4** with the IC_50_ value of 17.4 µg/mL as compared to the standard drug doxorubicin having an IC_50_ value 1.2 µg/mL [118]. Osmania et al. (2022) synthesized a new pyrimidine-triazole derivatives and carried out studies on its anticancer effect. A total of 10 novel Pyrimidine-Triazole derivatives were synthesized in this study and they were evaluated against three cancer cell lines; A549, MCF-7, and NIH3T3. IC_50_ values were calculated at 24 h and 48 h (incubation time). Two compounds, namely compound **5a** and **5b,** were found to be potent anticancer agents. Compounds **5a** and **5b** have the IC_50_ values of 1.573 μM and 3.698 μM after 48 h on MCF-7 cell lines respectively. The selective index of these compounds at the end of incubation period was 28.59 and 5.51, respectively. The control was Cisplatin with the IC_50_ value of 49.23 µM (48 hr) and Doxorubicin at 0.958 µM (48 hr) against the MCF cell line, respectively. The aromatase enzyme inhibition effects of the compound **5a** and **5b** is also calculated having the IC_50_ values of 0.082 and 0.198 μM [119]. Qin et al. (2015) synthesized fifteen 2,4-diaminopyrimidines and evaluated their biological activity as selective Aurora A kinase inhibitors. Their cytotoxic activity was tested against five cell lines. The most potent compound reported in this study was compound **6** with IC_50_ values of 3.6 µM (HCT-8), 0.5 µM (A-549), 0.9 µM (HeLa), and 2.4 µM (Hep-G2). VX680 was used as the standard, having IC_50_ of 44.6 µM (HCT-8), 19.4 µM (A-549), 27.3 µM (HeLa), and 63.4 µM (Hep-G2). Molecular docking studies revealed that compound **6** formed a major interaction with Aurora A, which showed 35-fold greater selectivity for Aurora A than for Aurora B. Additionally, compound **6** caused HeLa cells to arrest in the G2/M cell cycle. The IC_50_ values of compound **6** were 0.012 and 0.043 µM against Aurora A and Aurora B respectively whereas that of standard VX-680 were 0.261 and 0.453 µM respectively [120]. Filho et al. (2021) synthesized 6-ferrocene/heterocycle-2-aminopyrimidine and 5-ferrocene-1H-Pyrazole derivatives by microwave-assisted Atwal reaction and carried out docking, machine learning, and anti-proliferative activity studies of these agents. The compound **7** showed the potent anticancer activity. It was tested on 4 cancer cell lines like HCT116, PC3, HL60, and SNB19 with IC_50_ values of 56.99, 33.56, 70.26, and 85.11 µM respectively. The docking studies revealed that the compound **7** is the most active compound having the binding energy of −6.3 Kcal/mol with targeted protein (PDB:4HLW) [121].

El-Deen et al. (2022) designed and synthesized pyridothienopyrimidine derivatives and evaluated their anticancer and antimicrobial activity. The cytotoxic activity of newly synthesized compounds was evaluated against three cell lines, MCF-7, HepG-2, and WISH. The compounds that showed potent activity were examined as EGFR kinase inhibitors. The most potent anticancer agent reported in this study was compound **8** with IC_50_ values of 1.17, 1.52, and 417.55 µM against the cell lines HepG-2, MCF-7, and WISH, respectively. It is compared with the control drug Doxorubicin with IC_50_ values of 2.85, 3.58, and 432.10 µM against the cell lines HepG-2, MCF-7, and WISH, respectively. In terms of in vitro enzymatic inhibitory activity against EGFR kinase, compound **8** showed IC_50_ of 7.27 nM as compared to control Erlotinib with IC_50_ value of 27.01 nM [122]. Al-Anazi et al. (2022) synthesized pyrazoline and pyrimidine derivatives as novel epidermal growth factor receptor (EGFR) inhibitors and tested their anticancer property. All the synthesized compounds showed good anticancer activity. Among them the compound **9** showed the most potent activity against MCF-7 cell line with the IC_50_ value of 5.5 µM and selective index 18.18 as compared to tamoxifen standard with IC_50_ value of 26.95 µM [123].

El-Metwally et al. (2021) synthesized thieno pyrimidine-based derivatives as potent VEGFR-2 kinase inhibitors and evaluated their anti-cancer properties using MTT against the HepG2, HCT-116, and MCF-7 human cancer cell lines using sorafenib as a positive control. The most potent derivative reported was compound **10** with IC_50_ value 0.23 µM against the MCF-7 cell line which is similar to the control sorafenib with IC_50_ value 0.23 µM [124].

Madia et al. (2021) designed and synthesized pyrimidine-based analogues and evaluated their anticancer activity against HT-29, U-87 MG, MDA-MB231, CAL27, and FaDu cancer cell lines. Each of synthesized compounds showed excellent anticancer activity. Among them, compound **11** showed excellent activity at 24 and 48 h time intervals with EC_50_ values of 10.2, 22.0, 18.0, 9.7, and 26.2 and 5.4, 7.5, 7.9, 4.3, and 8.5 µM, respectively with control RDS 3442 (EC_50_ values of 51.8, 75.2, 34.8, 54.2, and 69.3 at 48 h respectively) [125].

All the reported pyrimidine derivatives (Figure 3 (**1**–**11**)) were acting as anticancer agents. The most potent compound among them was compound **10**, having the lowest IC_50_ value of 0.23 µM against MCF-7 cell line. It has a hydrazine moiety joined by phenyl amino linkage and pyrimidine with benzothiophene nucleus which may be responsible for its excellent activity.

### 3.2. Quinoline Derivatives as Anticancer Agents

Quinoline is a nitrogen containing heterocyclic aromatic compound, also known as 1-aza-napthalene or benzopyridine. Its molecular weight is 129.16 and molecular formula is C_9_H_7_N. The log P value is 2.04 while the pKb and pKa values are 4.85 and 9.5, respectively. Quinoline is a weak tertiary base and with acids it can produce salt and exhibits reactions akin to those of pyridine and benzene. Numerous naturally occurring chemicals (Cinchona Alkaloids) and pharmacologically active molecules with a wide range of biological activities include the quinoline nucleus in their structure. Diverse pharmacological activities (anticonvulsant, analgesic, cardiotonic, antibacterial, antifungal, anti-inflammatory, and anti-malarial) of quinoline have been reported [126].

Hamdy et al. (2019) conducted anticancer study on quinoline based heterocycles targeting Bcl-2. Out of these, Compound **12** showed excellent activity in MDA-MB-231, HeLa, KG1a, and Jurkat with the IC_50_ values of 0.54, 1.42,1.21, and >100 µM respectively. This compound showed the IC_50_ value of 0.15 µM against Bcl-2 compared with Gossypol (IC_50_ value of 0.60 µM). Compound **12** had sub-micromolar anti-proliferative activity in cancer cell lines that express Bcl-2, as well as a sub-micromolar IC_50_ value in an ELISA experiment using the Bcl2-Bim peptide [127]. Mathada et al. (2022) studied the anticancer effect of quinoline and its derivatives. Hence, 62 compounds were synthesized in this study. Out of these, the most potent compound was found to be compound **13**, which showed a potent anticancer effect against three cell lines, MDA-MB-231, HeLa, and SMMC-7721 with IC_50_ values of 0.12, 0.08, and 0.34 µM respectively. Etoposide was used as standard drug with IC_50_ values of 5.26, 2.98, and 3.48 µM respectively. Compound **13** had a unique capacity to induce apoptosis in HeLa cells, halt the cell cycle at the G0/G1 phase, elevate intracellular ROS (Reactive Oxygen Species) levels, and decrease mitochondrial membrane potential. Additionally, it had the ability to significantly reduce MEK1 kinase activity and disrupt the Ras/Raf/MEK/ERK transduction pathway [128].

Katariya et al. (2020) performed the studies on anticancer, antimicrobial activities of quinoline based hydrazone analogues. Nine compounds out of all the tested compounds showed significant anti-cancer activity at 10 µM. These compounds were then screened at 10-fold dilutions of five different concentrations (0.01, 0.1, 1, 1, and 100 µM) with GI_50_ values ranging from 0.33 to 4.87 µM and LC_50_ values ranging from 4.67 µM to >100 µM. The most active molecule, compound **13,** had a mean graph midpoint (MG-MID) GI_50_ value of 1.58 µM, which was significantly lower than commercially available standards Bendamustine and chlorambucil (60 and 52 µM, respectively). The IC_50_ value of compound **14** was 132 µM against the NIH/3T3 cell line, which was significantly higher than the GI_50_ values of compound **14** against the NCI 60 cancer cell lines, which demonstrated that compound **14** was highly selective for cancer cells at concentrations that are lower than the effective concentration against healthy cell lines [129].

George et al. (2019) synthesized the new derivatives of quinoline, i.e., 4,5- dihydropyrazoles as EGFR inhibitors and evaluated their anti-proliferative activity. The newly synthesized compounds were tested against HeLa, MCF-7, and DLD1 cancer cell lines as well as normal fibroblast WI-38. Eight compounds showed potent activity towards the DLD1 cell line and it was safe to normal cell lines. The most active compound in this study was compound **15**, with IC_50_ values of 0.227, 0.136, 1.277, and >94.5 µM against the cell lines MCF-7, HeLa, DLD1, and WI-38, and it was compared to the control drug CHS 828 with the IC_50_ values of 0.018, 0.040, 2.315, and >134.71 µM respectively. Compound **15** showed the EGFR inhibitory activity with IC_50_ value of 31.8 nM as compared to the control Gefitinib 29.16 nM [130]. Koprulu et al. (2021) had conducted anticancer activity and molecular docking studies of quinoline derivatives. The cytotoxic activity of synthesized piperazine substituted quinoline derivatives were tested against three cell lines, namely rat glioblastoma (C6), human cervical cancer (HeLa), and human adenocarcinoma (HT29). The docking studies revealed that compound **16** was the most potent compound for metastatic cancer treatment due to its binding affinity to PLCγ1. The IC_50_ values of compound **16** were 100, 144.8, and 117.6 µM against the cell lines, C6, HeLa, and HT29. Further, 5-Fluro Uracil was used as the standard drug with IC_50_ values of 163, 469.6, and 501.2 µM [131].

Ramya et al. (2018) synthesized and evaluated anticancer potential of curcumin inspired 2-chloro/phenoxy quinoline analogues. This study described the synthesis of twenty novel 2-chloro/phenoxyquinoline derivatives inspired by curcumin. The cytotoxic activity of the acquired novel chemical entities toward various tumor cell lines was examined in vitro. The most active compound **17** showed potential cytotoxicity against different cell lines HeLa, HGC, NCI-H460, DU-145, PC-3, and 4T1 with IC_50_ values of 5.41, 8.73, 3.96, 3.99, 3.12, and 1.81 µM respectively. The reference drug curcumin had the IC_50_ values of 17.11, 33.15, 18.65, and 18.65 µM against the cell lines NCI-H460, DU-145, PC-3, and 4T1, respectively. Additionally, study of ROS levels, DAPI staining, AO-EB labelling, and annexin binding assay showed that the promising compound **16** may cause apoptosis in PC-3 cells and G2/M cell cycle arrest [132].

Upadhyay et al. (2018) conducted the synthesis and screening of pyrano[3,2-c],quinolones derivatives. One pot multicomponent condensation was carried out between malononitrile, 2,4-dihydroxy-1-methylquinoline, and substituted aromatic aldehydes to create a number of pyrano[3,2-c]quinoline based structural analogues. The compounds were accessed for their cytotoxic and anti-inflammatory activity. Out of these, compound **18** was found to be the most potent anticancer agent and showed 81–53% anti-proliferative inhibition at 1 µM concentration against all cell lines, including ACHN, Panc-1, HCT-116, H-460, and Calu-1, with IC_50_ values of 0.1, 1.0, 0.3, 0.3, and 0.5 µM respectively. Flavopiridol, having the IC_50_ values of 71, 78, 71, 88, and 74 µM, and gemcitabine, with 73, 74, 73, 71, and 79 µM, respectively, were used as standard [133].

Hagras et al. (2021) designed, synthesized, and conducted docking studies and anti-proliferative evaluation of newly discovered quinolines. The most effective anti-tubulin polymerization agents are colchicine binding site inhibitors. In order to exhibit the same fundamental pharmacophoric characteristics as colchicine binding site inhibitors, novel quinoline compounds have been developed and synthesized. Using colchicine as a positive control, the synthesized compounds were evaluated in vitro against three human cancer cell lines; MCF-7, HepG-2, and HCT-116. The most potent compound **19** have the IC_50_ values of 1.89, 1.43, and 4.21 µM against the cell lines HePG2, HCT-116, and MCF-7 respectively. The control drug colchicine with IC_50_ values of 7.4, 9.3, and 10.4 µM, respectively, and its effect on cell cycle distribution were evaluated. The outcomes showed that compound **19** had the ability to stop the cell cycle at the G2/M phase. A double-staining test with annexin V and PI was performed to investigate the apoptotic effect of the synthesized compounds. Compound **19** caused HepG-2 cells to apoptosis thirteen times more frequently than control cells [134]. Mirzaei et al. (2019) synthesized novel quinoline chalcone hybrids and conducted studies on the structure–activity relationship and docking studies for their anticancer potential as active tubulin inhibitors. Four different human cancer cell lines, including A2780/RCIS (Cisplatin-resistant human ovarian carcinoma), A2780 (human ovarian carcinoma), normal Huvec cells, MCF-7 (human breast cancer cells), and MCF-7/MX (Mitoxantrone-resistant human breast cancer cells), were used to test the cytotoxic activity of synthesized substances. The most potent compound reported in this study was compound **20** with IC_50_ values of 2.32, 2.615, 4.96, 2.32, and 4.44 µM against the cell lines A2780, A2780/RCIS, MCF-7, MCF-7/MX, HUVEC respectively. It is compared with the standard drug CA-4 with IC_50_ values of 0.24, 0.22, 0.43, and 1.49 µM against the cell lines A2780, A2780/RCIS, MCF-7, and MCF-7/MX, respectively [135]. Chate et al. (2018) investigated novel spiro-pyrimido[5,4-b]quinoline-10,50-pyrrolo[2,3-d]pyrimidine] derivatives as promising anticancer agents. By using the MTT assay, the newly synthesized compounds’ anticancer effects were assessed in vitro against the A431, PC-3, MCF-7, and MCF-10A cancer cell lines. Out of these, the most potent was compound **21,** having an IC_50_ value of 36.25 µM for MCF-10A (normal breast epithelial cell lines), indicating five times more selectivity for MCF-7 cancer cells with the IC_50_ value of 7.82 µM. Sunitinib was used as a control having IC_50_ values of 6.48, 7.25, 20.66, and 34.55 µM against the cancer cell lines A-41, MCF-7, PC-3, and MCF-10A, respectively [136].

While evaluating the above reported quinoline derivatives (Figure 4 (**12**–**21**)), most of the compounds were evaluated with their cytotoxic activity commonly against HeLa, MCF-7, and HCT-116 cells. Compounds **13**, **15**, and **18** were reported to be potent anticancer compounds but the most potent one among them was compound **13** with the lowest IC_50_ value of 0.08 µM on HeLa cells, 0.12 µM on MDA-MB-231, and 0.34 µM on SMMC-7721, having a steroid nucleus directly attached with quinoline along with a hydrazine chain. This also demonstrated the ability to cause apoptosis in HeLa cells.

### 3.3. Carbazole Derivatives as Anticancer Agents

A polycyclic aromatic hydrocarbon called carbazole has a broad aromatic system and a central nitrogen atom that exhibits substantial electron delocalization. It consists of a five-membered nitrogen-containing ring sandwiched between two six-membered benzene rings. It has an indole-like structure, but at the indole position 2–3, a second benzene ring is fused to the five-membered ring. When creating electron donor-electron acceptor (D-A) chemical dyes, carbazole is frequently used as a conjugated bridge [137].

Murali et al. (2017) studied the creation of hetero annulated isoxazolo-, pyrido-, and pyrimido carbazoles, also screening them for in vitro anticancer activity. By cyclo condensation with the appropriate reactants (hydroxylamine hydrochloride, malononitrile, and guanidine nitrate), the newly synthesized heterocycles isoxazolo-, pyrido-, and pyrimidocarbazoles were produced from the readily available 2-(3′-bromo-4′-methoxybenzylidene)-2,3,4,9-tetrahydro-1H-carbazol. All the synthesized substances were tested for in vitro cytotoxicity against the A-549 and MCF-7 human cancer cell lines. Compound **22** demonstrated substantial activity against MCF-7 with the IC_50_ value of 20 µM as compared to cisplatin IC_50_ value of 18 µM. All other compounds showed moderate to powerful activity and consequent apoptotic cell death, which was demonstrated by AO/EB and DAPI of fluorescence microscopy analysis [138]. Wang et al. (2011) designed and synthesized substituted 11H-benzo[a] carbazole-5-carboxamides as novel anticancer agents and tested them against human cancer A549 and HCT-116 cell lines. The most potent derivative was compound **23** with IC_50_ value 8.2, 9.5 µM against the cancer cell lines A549, and HCT-116 respectively. Amonafide was used as the standard with IC_50_ values 8.1 µM and 15.3 µM, respectively [139]. Debray et al. (2010) synthesized N-ethoxycarbonyl-N-arylguanidines by the montmorillonite K-10 catalyzed cyclization, which provides access to pyrimido[4,5-c]carbazole and pyrimidoo[5,4-b]indole derivatives. Further, 3-aminocarbazole and 3-aminoindole were converted into two novel heterocycles, pyrimido[4,5-c]carbazole and pyrimido[5,4-b]indole, respectively. With the aid of montmorillonite K-10 clay as a catalyst and microwave irradiation, the essential Friedel–Crafts intramolecular cyclization was accomplished. The micromolar IC_50_ of the pyrimido[4,5-c]carbazole derivative was significant against cancer cell lines. The most potent compound in this study was found to be compound **24** with the IC_50_ values 17 and 17 µM against the cancer cell lines HL60 N and HL60 MX2, respectively. Etoposide is used as a reference with IC_50_ values of 1.3 µM and 11.9 µM [140].

Sun et al. (2017) synthesized novel carbazole sulfonamide derivatives and evaluated their anti-proliferative activity and aqueous solubility as an antitumor agent. A number of novel carbazole sulfonamide derivatives were produced by the current optimization of IG-105 on the carbazole-ring. Each compound’s anti-proliferative effectiveness was tested on HepG2 cells. When tested for anti-proliferative activity against MIA PaCa-2 (pancreatic cancer), MCF-7 (breast cancer), and Bel-7402 (hepatocellular/liver cancer), compounds that had shown action superior or equivalent to that of IG-105 against HepG2 were found to be effective. Five of the seven substances were chosen for further investigation and discovered to have IC_50_ values against the four cell lines that were comparable to those for IG-105. The activity of two compounds, compound **25a** and **25b** against HepG2 and MCF-7 (IC_50_ values of 0.01 and 0.07 µM) was close to that of the positive controls podophyllotoxin and CA-4. The most potent compound out of these two was Compound **25a** with IC_50_ value against HEPG2 was 0.012 µM as compared to the controls podophyllotoxin with IC_50_ value 0.003 µM and CA-4 with IC_50_ value of 0.002 µM respectively [141]. Arya et al. (2018) performed eco-compatible synthesis of highly functionalized pyrido[2,3-a] carbazole derivatives and tested them for their cytotoxic effects on cancer cell lines MCF-7 and A549. The result of this study revealed compound **26** as the potent anticancer compound with lowest IC_50_ values of 45 µM and 50 µM against MCF-7 and A549 while the control drug Ellipticine presented IC_50_ values of 73 µM and 65 µM, respectively [142]. Padmaja et al. (2014) tested the anticancer activity of novel pyrano[3,2-c]carbazole derivatives which induce cell death by the inhibition of tubulin polymerization. A facile one-pot, three-component reaction using malononitrile-ethyl cyanoacetate, aromatic aldehydes, and 4-hydroxycarbazoles, catalyzed by trimethylamine was used to synthesize pyrano[3,2-c]carbazole derivatives. Investigations were performed to test their ability to inhibit the proliferation of several cancer cell lines, including K562, MDA-MB-231, HeLa, and A549. The most potent anticancer agent was compound **27** with IC_50_ values of 0.43, 1.13, 3.41, 6.12, and 69.24 μM against the cell lines MDA-MB 231, K562, A549, HeLa, and L929, respectively. Combrestatin A-4 (CA-4) was used as the control. This compound causes apoptosis by preventing tubulin polymerization and G2/M phase arrest of the cell cycle [143]. Patel et al. (2021) synthesized and characterized coumarin carbazole based functionalized pyrimidines, evaluated the anticancer effect, and performed molecular docking studies. The synthesized compounds were evaluated against three cell lines (HeLa, NCI-H520, and NRK-52E). Compound **28a** and **28b** were reported to be most active because of their ability to cause apoptosis and cell cycle arrest. Both of these molecules demonstrated high binding affinity towards CDK2 protein. Compound **28a** was the most potent anticancer agent with IC_50_ values of 12.59 µM, 11.26 µM, and 28.37 µM against the cell lines HeLa, NCI-H520, and NRK-52E respectively. It was compared to the control Cisplatin with IC_50_ values of 7.75, 10.41, and 12.93 µM and 5-Flurouracil with IC_50_ values of 55.72, 8.36, and 46.68 µM, respectively [144]. Huang et al. (2021) performed the synthesis of novel carbazole derivatives as selective and potent anticancer drugs, as well as performed their biological evaluation and structure–activity relationship studies. The in vitro cytotoxic effects of two series of carbazole compounds against the three cell lines A875, HepG2, and MARC145 were assessed. In this study, compared to the control 5-fluorouracil, the results showed that some of these carbazole derivatives had much better cytotoxic effects against the examined cell lines. Particularly, the carbazole acylhydrazone compounds **29a** and **29b** showed strong inhibitory effect against cancer cells, but essentially no activity against normal cells. Particularly, compound **29a** showed significantly selective proliferation inhibition on cancer and normal cell lines with the selectivity index up to 13, and it had a stronger inhibitory effect on the A875 with IC_50_ value of 7.65 µM and HepG2 cell lines with IC_50_ value of 8.16 µM compared to the normal cells line MARC145 with IC_50_ value of > 105 µM. The control 5-Flurouracil had IC_50_ values of 72.33, 81.94, and 77.56 µM against A875, HepG2, and MARC145 cell lines, respectively [145].

Chen et al. (2018) synthesized racemic and chiral carbazole aminoalcohols and tested their anticancer potential. It was found that the topoisomerase I was inhibited due to a number of reasons, e.g., substitution site, heterocycle, chirality, and length of alkyl chain. MDA-MB-231, HCT116, and A549 cancer cell lines were used as test subjects and carbazole amino alcohols with potent topo I inhibitory activities, such as pyrrolidine derivative and propyl- to pentyl- amine derivatives, demonstrated good anticancer activities. The most potent compound reported in this study was compound **30** with broad spectrum anti-tumor activity against 15 cancer cell lines. The IC_50_ values were 7.9, 4.6, 2.8, 7.5, 3.9, 3.4, 3.8, 2, 4.8, 1.6, 1.5, 1.7, 1.7, 1.7, and 2.2 μM against A549, HCT116, MDA-MB-231, HeLa, H3122, BT549, BEL7402, 3AO, HO-8910, Rh30, Pfeiffer, Molm13, OC1-AML2, Jurkat, and HL60 cell lines, respectively [35].

Vairavelu et al. (2014) performed solvent-free synthesis of heteroannulated carbazoles using grinding conditions and a number of new carbazole analogues. Templates for isoxazolo, pyrido, pyrazolo, and pyrimido were developed and synthesized in high yield. The synthesized compounds were tested in vitro for their anticancer potential. Compound **31** was the most potent anticancer compound with IC_50_ values of 0.37 µM and 15.12 µM against HeLa and AGSb cell lines respectively. Ellipticine was used as the control with IC_50_ values of 4.12 and 7.33 µM, respectively [146].

Among the above-mentioned carbazole derivatives (Figure 5 (**22**–**31**)), the most potent one was compound **25a** having a pyridine ring joined by sulphonamide linkage and substituted with 2,5-dimethoxy group, having the lowest IC_50_ value of 0.012 µM against the HEPG2 as compared to the control podophyllotoxin with IC_50_ value of 0.003 µM and CA-4 with IC_50_ value of 0.002 µM.

### 3.4. Pyridine Derivatives as Anti-Cancer Agents

Pyridine has the chemical formula C_5_H_5_N and is a fundamental heterocyclic organic molecule. The word “pyridine” is taken from Greek and combines the words “idine” and “pyr,” which both refer to aromatic bases. Picoline, the first pyridine base, was discovered by Anderson in 1846. It took quite some time for Wilhelm Korner and James Dewar to discover its structure. It resembles the well-known and fundamental aromatic molecule benzene in many ways, but with one C-H group replaced by an atom of Nitrogen. Like benzene, pyridine possesses a conjugated system of six delocalized electrons distributed around the heterocyclic ring. The molecule satisfies the Huckel requirements for aromaticity and is planar in nature [147].

Gomha et al. (2018) performed the studies on anti-tumor activity through the Synthesis of some new pyridine-based heterocyclic compounds. By the reaction of 7-(pyridin-4-yl)-2-thioxo-2,3-dihydropyrido[2,3-d] pyrimidin-4(1H)-one and 2-cyano-N-(1-(pyridin-4-yl)ethylidene)-acetohydrazide with hydrazonoyl halides, 5-amino-N-(1-(pyridin-4-yl)ethylidene)-1H-pyrazole-4-carbohydrazides and 8-(pyridine-4-yl)pyrido[2,3-d][1,2,4]triazolo[4,3-a]pyrimidin-5(1H)-ones were synthesized. The anti-cancer activity of the synthesized compounds was tested against the HEPG2 cell line. The most potent compound reported in this study was compound **32**, with an IC_50_ value of 0.97 µM against the HEPG2 cell line Doxorubicin is used as the standard with an IC_50_ value of 0.74 µM [148]. Fayed et al. (2019) designed, synthesized, and performed molecular modeling studies of coumarin derivatives. The synthesized compounds were evaluated for their cytotoxic activity against different cell lines; HCT-116, MCF-7, A549, and HepG-2. The most potent compound in this study was compound **33** with IC_50_ values of 1.11, 6.44, 4.51, and 7.18 µM against the cancer cell lines MCF-7, HCT-116, HepG-2, and A549, respectively. 5-Flurouracil was the standard with IC_50_ values of 7.76, 8.78, 8.15, and 7.65 µM, respectively. Compound **33** induced cell cycle arrest in the G2/M phase and apoptosis. Caspase-3 is used for stimulating apoptosis [149]. Nagender et al. (2016) synthesized novel hydrazone and azole-functionalized pyrazolo[3,4-b]pyridine derivatives. With the key intermediate ethyl 2-(3-amino-6-(trifluoromethyl)-1H-pyrazolo[3,4-b]pyridin-1-yl)acetate, a variety of pyrazolo[3,4-b]pyridine compounds were synthesized. The reaction was carried out with hydrazine hydrate followed by isothiocyanate, acid chloride, and different aldehydes to form 1,2,4 triazoles, oxadiazoles, hydrazones, and thiadiazoles. All the synthesized compounds were screened for anti-cancer activity. With an IC_50_ value of 3.2 µM, compound **34** with the trifluoromethylthio group demonstrated excellent action against the lung cancer cell line (A549) compared with the control 5-Fluorouracil with an IC_50_ value of 1.7 µM [150].

El-Naggar et al. (2018) conducted the synthesis and biological evaluation of pyridine-urea derivatives as anti-cancer drugs. All the synthesized compounds were tested for their in vitro antiproliferative activity in breast cancer against the MCF-7 cell line. Two compounds, **35a** with IC_50_ values of 0.22 µM (48 h) and 1.88 µM (72 h) and **35b** with IC_50_ values of 0.11 µM (48 h) and 0.80 µM (72 h), were reported to present potent anti-cancer activity against the MCF-7 cell line. The most potent compound was compound **35b** with the lowest IC_50_ value, while compared to the reference drugs, doxorubicin presented IC_50_ values of 1.93 µM (48 h) and 1.07 µM (72 h) and sorafenib with IC_50_ values of 4.50 µM (48 h) and 1.71 µM (72 h) respectively [151].

Dinda et al. (2014) carried out a study on the cytotoxicity of pyridine-wingtip substituted annelated N-heterocyclic carbene complexes of silver (I), gold (I), and gold (III). Three new compounds were synthesized from 1-methyl-2-pyridin-2-yl-2H-imidazo[1,5-a]pyridin-4-ylium chloride. The synthesized complexes were tested for their cytotoxicity against HepG2 (human hepatocellular carcinoma), A549 (human lung adenocarcinoma), HCT 116 (human colorectal carcinoma), and MCF-7 (human breast adenocarcinoma) cells. Compound **36** was the most potent antiproliferative agent with IC_50_ values of 4.91, 5.08, 5.23, and 5.18 µM against the cell lines HepG2, HCT-116, A549, and MCF-7, respectively. Cisplatin was used here with IC_50_ values of 4.31, 4.89, 6.12, and 4.42 µM, respectively [152]. El-Gohary et al. (2019) synthesized novel pyrazolo[3, 4-b]pyridine analogs, screened their anti-cancer potential, and performed molecular modeling. An in vitro anti-cancer assay toward MCF-7, HepG2, and Hela cancer cells and in vivo anti-cancer assay over E.A.C. in mice were carried out. The most potent compound was compound **37**, with IC_50_ values of 3.63, 3.11, and 4.91 µM against the cell lines HepG2, MCF-7, and Hela, respectively. Doxorubicin was used as the reference with IC_50_ values of 4.3, 3.97, and 5.17 µM, respectively. According to molecular modeling studies, it was reported that the synthesized compounds bind to DNA through intercalation similar to doxorubicin [153].

Sangani et al. (2014) designed, synthesized, and evaluated the molecular modeling of pyrazole-quinoline-pyridine hybrids as a potential class of antibacterial and anti-cancer drugs. A one-pot multicomponent reaction was carried out using a base-catalyzed cyclo condensation reaction, and a new series of pyrazole-quinoline-pyridine hybrids were synthesized. All the compounds synthesized tested for in-vitro anti-cancer and antimicrobial activity. The most potent compound with the lowest cytotoxicity was compound **38**, compared to reference drug Erlotinib with IC_50_ values of 0.13, 0.12, and 0.032 µM, against the cell lines A549, HepG2, and E.G.F.R. respectively. Compound **38** showed three hydrogen bonds and one -π cation interaction with a minimum binding energy of −54.6913 kcal/mol in the molecular docking studies with the catalytic pocket of protein [154]. Elzahabi (2011) synthesized and characterized some benzazoles bearing pyridine moiety in a search for novel anti-cancer agents. Thirteen new benzazole compounds were synthesized as potential anti-cancer agents. The National Cancer Institute (NCI), U.S.A. chose four derivatives of the produced compounds to be tested for their anti-cancer potential against a panel of 60 cancer cell lines at a single high dose. Compounds 4-[p-chlorophenyl]pyridine and 4-[pmethoxyphenyl]pyridine were chosen for further testing at five doses after exhibiting a broad and moderate anti-cancer activity against 41 tumor cell lines belonging to the nine subpanels employed. The compounds **39a** and **39b** showed excellent activity in HOP-92 cells with GI_50_ values of 0.275 µM and 2.65 µM, respectively. Regarding colon cancer, compound **39a**’s GI_50_ value against HCT116 was 3.47 µM, and the most sensitive cell line in this subpanel was KM12 towards compound **39b** with GI_50_ 4.79 µM. The most sensitive lines for compounds **39a** and **39b** were SF-539 and SNB-75, belonging to C.N.S. cancer cell lines with GI_50_ 2.07 µM and 2.36 µM, respectively [155].

Zheng (2014) researched the design, synthesis, and biological evaluation of novel combretastatin-A4 pyridine-bridged analogs as anti-cancer drugs. The synthesized compounds potentially suppressed cell growth and survival, stopped the cell cycle, and prevented angiogenesis and the development of blood vessels, similar to CA-4 (combretastatin-A4). The IC_50_ values of compound **40a** against three cancer cell lines were 0.0031, 0.089, and 0.0038 µM against MDA-MB-231, A549, and HeLa, respectively, and for compound **40b** were 0.0046, 0.044, and 0.0014 µM respectively. The reference compound used was combretastatin-A4 with IC_50_ values of 0.0028, 0.0038, and 0.0009 µM, respectively [156]. Abbas et al. (2015) synthesized novel pyridines having an imidazole moiety. The one-pot multicomponent reaction of 5-acetyl imidazole, substituted benzaldehyde (or terephthaldehyde), malonitrile (or ethyl cyanoacetate or diethyl malonate), and ammonium acetate was carried out to produce a novel series of pyridine and bipyridine derivatives. Some recently synthesized compounds were tested for their anti-cancer activities against the human breast cell line (MCF-7) and liver carcinoma cell line (HEPG2) with the reference Doxorubicin IC_50_ values of 0.46 µM and 0.42 µM, respectively. The most potent compound was compound **41**, with IC_50_ values of 1.7 and 6 µM, respectively [157]. Ivasechko (2022) synthesized novel pyridine-thiazole hybrid molecules as potential anti-cancer agents and evaluated their anti-cancer activity against several kinds of tumors, e.g., carcinomas of the breast, lung, colon, glioblastoma, and leukemia. Two compounds showed the highest anti-cancer activity. The most potent compound reported was compound **42**, with an IC_50_ value of 2.79 µM against the MCF-7 cell line. Doxorubicin was standard with an IC_50_ value of 1.04 µM against the MCF-7 cell line [158].

Among the above-reported potent anti-cancer compounds (Figure 6 (**32**–**42**)) having a pyridine moiety, the most active compound which showed the best cytotoxic activity was compound **40a** (a diaryl compound joined by pyridine ring. Diaryl rings were substituted with methoxy group on different positions). It showed the lowest IC_50_ value of 0.0031 µM, 0.089 µM, and 0.0038 µM against the three human cancer cell lines MDA-MB-23, A549, and HeLa, respectively.

### 3.5. Imidazole Derivatives as Anticancer Agents

Heinrich Debus synthesized the first Imidazole in 1858 by reacting glyoxal and formaldehyde in the presence of ammonia. Derivatives of Imidazole are now widely used in many treatments and have recently attracted much attention. Various medicinal properties of imidazole-based hybrid molecules have been reported, especially as antitumor, anti-diabetic, anti-HIV, anti-mycobacterial, anti-inflammatory, analgesic, and anti-protozoal activities [159].

Liu et al. (2015) synthesized carbazole-imidazole-based carbazole derivatives and tested them for anticancer activity against various cell lines: HL-60, SMMC-7721, A549, MCF-7, and SW480. Compound **43** demonstrated potent activity among the synthesized carbazole derivatives, with IC_50_ values of 0.51, 2.38, 3.12, 1.40, and 2.48 μM, whereas control cisplatin (DDP) had IC_50_ values of 1.32, 6.24, 11.83, 15.17, and 12.95 μM, respectively. Furthermore, Compound **43** caused cell cycle arrest in SMMC-7721 cells at IC_50_ of 0.51 µM [160].

Ruzi et al. (2021) synthesized imidazole derivatives having ethyl 5-amino-1-N-substituted imidazole-4-carboxylate building blocks as anticancer agents. Compound **44** showed the best anticancer activity with HeLa, HT-29, A549, HCT-15, and MDA-MB-231 cell lines with IC_50_ values of 0.81, 1.77, 15.22, 17.92, and 5.48 μM. Reference drug Doxorubicin presented IC_50_ values of 0.1, 0.03, 1.1, 0.27, and 0.51 Μm, respectively [161]. Singh et al. (2021) synthesized and reported C_6_-substituted benz[4,5]imidazo[1,2-α]quinoxaline derivatives and performed their anticancer evaluation. Compound **45** showed excellent anticancer activity at IC_50_ values of 14.62, 4.78, 1.55, and 1.04 μM against MDA-MB-231, MDA-MB-486, MCF-7, and MCF12A cell lines, respectively, whereas control cisplatin had IC_50_ values 3.26, 1.43, and 4.23 μM against MDA-MB-231, MDA-MB-486, and MCF-7 cell lines, respectively [162]. Yang et al. (2012) synthesized various novel hybrids of 2-phenylbenzofuran and Imidazole. Among them, compound **46** displayed good in vitro activity against different cancer cell lines, including SMMC-7721, SW480, MCF-7, A549, and HL-60, with IC_50_ values of 1.65, 3.38, 5.87, 10.93, and 2.49 μM, respectively, whereas control cisplatin (DPP) had IC_50_ values of 8.86, 15.92, 1.65, 11.68, and 1.81 μM respectively. Compound **46** demonstrated good anticancer activity with more excellent cytotoxic activity than standard DPP [163]. Song et al. (2012) synthesized different novel hybrids with 2-substituted benzofuran and imidazole. In vitro anticancer activities of synthesized hybrids were tested against a panel of human tumor cell lines, such as the ovarian carcinoma cell line (Skov-3), leukemia (HL-60), and breast carcinoma (MCF-7). Compound **46** was found to be more selective than standard DPP. Compound **47** showed more potent activity at IC_50_ values of 9.5, 8.4, and 11.8 µM with standard drug Cisplatin (DDP) (IC_50_ values of 8.9, 5.5, and 13.0 µM respectively) [164].

The design, synthesis, and anti-proliferative evaluation of novel imidazole-thione linked benzotriazole derivatives against MCF-7, HL-60, HCT-116 and HUVEC cell lines were reported by Khayyat et al. (2021). Anticancer activity screening results revealed that compound **48** was the most promising anticancer agent having IC_50_ values of 3.57, 0.4, 2.63 and 118.9 µM, respectively. CA-4 was used as a reference drug with IC_50_ values of 0.58, 0.77, 0.24, and 13.6 µM, respectively[165]. Quinazoline-based imidazole derivatives were synthesized and screened for their anticancer activity against epidermal growth factor receptor (EGFR) and HT-29 cell lines (normoxic and hypoxic conditions) by Kumar et al. (2022). Most of the synthesized compounds displayed potent anticancer activity. Among them, compound **49** showed excellent activity with IC_50_ values 0.47 and 2.21 µM, respectively, with control gefitinib IC_50_ values 0.45 µM against EGFR, and 3.63 µM and 5.21 µM against Normoxia and Hypoxia in HT-29 cells, respectively [166]. Kalra et al. (2021) synthesized imidazole and purine-derived derivatives as anticancer agents. These are tested for antiproliferative activity against different cancer cell lines, including MDA-MB-231, T47D, MCF-7, A549, and HT-29 by MTT assay. Compound **50** had IC_50_ values of 36.7, 9.96, 5.17, 2.29, and 3.29 µM, respectively, whereas control, Erlotinib, had IC_50_ values 5.46, 9.80, >30.00, 1.04, and 4.63 µM, respectively. Compound **50** showed the most potent anticancer activity against A549 cell lines with an IC_50_ value of 2.29 µM [167].

Taheri et al. (2020) synthesized and reported imidazole derivatives of anticancer importance. The imidazole derivatives showed a good cytotoxic effect on MCF-7, HT-29, and HeLa cell lines. The order of cytotoxic effect of the compounds on various cell lines was MCF7 > HT-29 > HeLa. Compound **51** showed the most potent activity at IC_50_ values of 2.5, 5.36 and 4.03 µM, respectively. Doxorubicin was used as positive control with the IC_50_ values 1.4 µM, 0.7 µM, and 0.9 µM respectively [168]. Oskuei et al. (2021) synthesized new imidazole chalcone derivatives and performed MTT assays against four cancer cell lines, including A549, MCF-7, HEPG2 and MCF-7/MX. Among them, compound **52** showed significant cytotoxicity with IC_50_ values of 7.05, 9.88, 20.2, and 3.86 µM with control combretastatin having IC_50_ values 0.86, 0.43, 0.63, and 1.49 µM respectively [169]. Wang et al. (2013) synthesized the novel 2-benzylbenzofuran and imidazole derivatives and compared their activity against various cancer lines, namely HL-60, A549, SW480, MCF-7, and SMMC-7721. Compound **53** showed potent cytotoxic activities among the synthesized molecules with IC_50_ values of 1.02, 3.57, 3.55, 2.29, and 3.09 µM with Control DDP (IC_50_ values of 3.10, 13.61, 12.32, 10.64 and 14.75 µM). Among the synthesized compounds, imidazole, 2-methyl-imidazole, or 2-ethyl-imidazole scaffolds showed weak cytotoxic activities. Compound **53** showed the selectivity towards the MCF-7 and SMMC-7721 with IC_50_ values of 1.02 and 3.09 µM) [170].

Du et al. (2021) synthesized and reported new series of 1,3,4-oxadiazole based imidazole based heterocyclic hybrids as anti-cancer agents. Among them, compound **54** showed the highest anticancer activity against HepG2, SGC-7901, and MCF-7 cell lines with IC_50_ values of 0.7, 30.0, and 18.3 µM. Standard drug, 5-Fluorouracil had IC_50_ values of 22.8, 28.9, and 16.7 µM respectively [171]. The above-reported anticancer compounds (Figure 7 (**43**–**54**)) have an imidazole moiety, which showed the best cytotoxic activity against representative cell lines. Furthermore, compound **49** showed the lowest IC_50_ value of 0.47 µM against epidermal growth factor receptor, whereas the control drug, gefitinib, had an IC_50_ value of 0.45 µM. Compound **49** had a quinazoline ring which was substituted with substituted aromatic ring joined by amino linkage.

### 3.6. Benzimidazole Derivatives as Anticancer Agents

Benzimidazole consists of benzene fused to the 4,5-positions of the imidazole ring. It is known as benzimidazole or benzoglyoxaline. The resonance in benzimidazole demonstrates its amphoteric nature and indicates that an electrophilic attack will occur either at N-1 or in the benzene ring [172]. Various studies have revealed that substituted benzimidazoles and heterocycles can easily interact with biopolymers, having pharmacological activity with lower toxicities. Because of the fused nitrogen nuclei, benzimidazoles are structural isosteres of nucleobases and readily interact with biomolecular targets, eliciting a wide range of biological activities, such as anti-inflammatory, antiulcer, anti-hypertensive, anthelmintic [173], and anticancer properties. For example, nocodazole, benzimidazole containing antineoplastic drug works by depolymerizing microtubules to achieve its effects [174].

Shao et al. (2014) synthesized and reported novel pyrimidine-benzimidazole derivatives with anticancer properties against MCF-7, MGC-803, EC-9706, and SMMC-7721 cancer cell lines. Compound **55** showed excellent anticancer activity against cancer cell lines with IC_50_ values of 1.40, 1.07, 2.79, and 19.28 µM respectively. The standard drug 5-flourouracil had IC_50_ values of 7.12, 3.45, 8.07, and 15.08 µM respectively [175]. Husain et al. (2012) designed and synthesized two series of compounds having benzimidazole rings merged with triazole-thiadiazole and trizolothizdiazole. The synthesized compounds showed good anticancer activity against CCRF-CEM, HL-60 (T.B.), MOLT-4, RPMI-8226, and S.R cell lines. Compound **56** showed good anticancer activities against leukemia cell lines having log_10_GI_50_, log_10_TGI, and log_10_LC_50_ values 6.58, 6.01, and 4.00 µM, respectively [176].

Sana et al. (2021) synthesized cinnamide derived pyrimidine-benzimidazole having growth inhibitory effects against five cancer cell lines, including A549, PC-3, HeLa, MDA-MB-231, and L132. Among the synthesized compounds, molecules **57** exhibited potent activity at IC_50_ values of 2.21, 3.15, 7.29, 5.71, and 69.25 µM, respectively, while the control Nocodazole had IC_50_ values 2.39, 1.96, 3.48, 2.13, and N.D. (for L132), respectively [177]. Sireesha et al. (2020) synthesized benzoxazole-linked β -carbolines by combining two anticancer fragments. Compound **58** showed the most promising anticancer activity against a panel of cell lines, MCF-7, A549, colo-205, and A2780 with IC_50_ values of 0.092, 0.72, 0.34, and 1.23 µM, respectively, while the control etoposide had IC_50_ values of 2.11, 3.08, 0.13, and 1.31 µM respectively [178]. Shi et al. (2014) synthesized and screened quinazoline-4-amines with benzimidazole analogs to have anticancer activity as dual inhibitors of c-Met and VEGRF-2. Compound **58** displayed the most potent inhibitory activity against c-Met and VEGFR-2 with IC_50_ values of 0.05 µM and 0.02 µM. Moreover, compound **59** showed potent activity against cancer cell lines, namely MCF-7 and Hep-G2, at IC_50_ of 1.5 and 8.7 µM, respectively. Furthermore, it also interacted at the ATP-binding site of c-Met and VEGFR-2 [179].

Romero-Castro et al. (2010) synthesized 2-aryl-5(6)-nitro1H-benzimidazole derivatives by one multicomponent reaction and evaluated their anticancer potential against K562, HL60, MCF7, MDA231, A549, HT29, K.B., and HACAT cancer cell lines. Compound **60** showed anticancer activity against representative cell lines at IC_50_ of 4.9, 2.1, 8.0, 4.0, 0.028, 1.8, 3.0, and 22.2 µM respectively whereas standard drug carboplatin had IC_50_ values of 1.5, 0.3, 8.9, 2.6, 0.05, 0.9, 4.1, and 0.6 µM respectively [180]. Sivaramakarthikeyan et al. (2020) synthesized benzimidazole-pyrazole hybrids and evaluated their anticancer activity against various cancer cell lines. Most of the synthesized compounds demonstrated potent anticancer activity. Compound **61** had potent anticancer activity against SW1900, AsPC1, and MRCS with IC_50_ values of 30.9, 32.8, and 80.0 µM, respectively, whereas control gemcitabine had IC_50_ values of 35.09, 39.27, and 54.17 µM, respectively [181].

Perin et al. (2021) synthesized a new series of acrylonitrile derivatives from aromatic aldehydes and N- substituted-2-cyanoethyl benzimidazole and tested their anticancer activity on various human cancer cell lines, such as Htert rpe-1, Capan-1, HCT-116, NCI-H460, DND-41, HL-60, K-562, MM.1S, and Z-138. Compound **62**, with both cyano and isobutyl substituents attached to the benzimidazole moiety, demonstrated good anticancer activity with IC_50_ values of 4.3, 0.3, 0.6, 0.4, 0.2, 0.3, 2.1, 1.5, and 0.4 µM, respectively, with standard docetaxel, which had IC_50_ values of 0.0553, 0.0088, 0.0017, 0.0024, 0.0125, 0.0072, 0.0152, 0.0118, and 0.0142 µM respectively. Furthermore, It was tested and found not to affect normal cells [182].

Wu et al. (2015) synthesized benzimidazole-2-substituted phenyl and pyridinepropyl ketene derivatives and tested their cytotoxicity on three cancer cell lines, i.e., MCF-7, HepG2, and HCT116. The synthesized compound **63** showed potent anticancer activity at IC_50_ values of 9.80, 0.75, and 0.03 µM, respectively, with control drug 5-Foorouracil with IC_50_ values of 222.6, 174.5, and 56.96 µM, respectively [183]. Holiyachi et al. (2016) synthesized coumarin benzimidazole hybrids from the 4-formylcoumarins by using N-sulphonation and N-methylation reactions. Among them, compound **64** showed excellent anticancer activity against HeLa and HT29 cell lines at LC_50_ and GI_50_ of 10, 20, 40, and 80 µM/mL, respectively [184]. Tahlan et al. (2018) synthesized heterocyclic 1H-benzimidazole derivatives and tested them for anticancer activity against a cancer cell line (HCT116) using the SRB method. Their results were similar to standard 5-fluorouracil. Compound **65** exhibited the anticancer activity at an IC_50_ value of 4.12 µM, with control 5-fluorouracil (IC_50_ value of 7.69 µM) [185]. Yuan et al. (2019) synthesized 6-amide-2-aryl benzoxazole/benzimidazole derivatives and tested them in vitro against tumor cells and by inhibiting VEGFR-2 kinase. Among them, compound **66** showed good potent anticancer activity. It showed good anticancer activity against VEGER-2, EGFR, and different cell lines, including; HUVEC, HepG2, A549, and MDA-MB-231 at IC_50_ values 102.8, 58.3, 1.47, 2.57, 73.81, and 57.32 µM with control sorafenib (IC_50_ values 103.6, 57.8, 5.35, 2.41, 7.62, and 17.38 µM, respectively) [186]. The above-reported anticancer compounds (Figure 8 (**55**–**66**)) had a benzimidazole moiety, which showed the best cytotoxic activity against representative cell lines. Furthermore, compound **59** (having the quinazoline ring and a substituted aromatic ring) showed the lowest IC_50_ value of 0.02 µM against VEGFR-2 with the control drug gefitinib.

### 3.7. Triazole Derivatives as Anticancer Agents

Triazole has been used in pharmaceuticals, agrochemicals, artificial materials, artificial acceptors, supramolecular ligands, and biomimetic catalysts. Heterocyclics have received particular attention due to their diverse pharmacological activities. The triazole ring is an essential five-membered heterocycle with three nitrogen atoms [187]. When creating new drug molecules, the triazole ring is a crucial isostere of imidazole, oxazole, pyrazole, and thiazole moieties. Many triazole-based derivatives have been extensively prepared and investigated for biological activities, which is one of the most active areas in new drug research and development [188]. Kurumurthy et al. (2019) synthesized pyrimidine based triazole derivatives and assessed their anticancer activity. Most of the synthesized compounds showed potent anticancer activity. Among them, compound **67** showed excellent activity against various cell lines, including U_937_, THP-1, and Colo205 cells, having IC_50_ values of 6.20, 11.27, and 15.01 µg/mL with control etoposide having IC_50_ values of 17.94, 2.16, and 7.24 µg/ML respectively [189].

Al-Blewi et al. (2021) designed and synthesized new imidazole derivatives with good anticancer properties based on the triazole pharmacophore. Four cancer cell lines, including Caco2, HTC116, HeLa, and MCF-7, were used for anticancer assessment. From all synthesized compounds, the recombinant click adducts triazole (compound **68**) was found to be the most potent with the IC_50_ values of 4.67, 16.78, 6.87, and 0.38 µM, respectively, with standard doxorubicin (IC_50_ values of 5.17, 5.64, 1.25 and 0.65 µM respectively). Furthermore, it showed excellent activity against MCF-7 cell line IC_50_ of 0.38 µM and was more potent than the control drug, doxorubicin (IC_50_ 0.65 µM) [190]. Bozorov et al. (2019) synthesized 1,2,3-triazole-linked isoxazole-benzothiazole-benzoxazole hybrids and tested them for anticancer activity against HeLa, A549, and HEK-293 cells. Compound **69** showed potent activity with IC_50_ values of 1.768 and 2.594 µM, and it was not toxic to normal cells. Compound **69** showed cell cycle arrest in the sub G1 cell cycle phase with 49.43% of cells. Moreover, compound **69** (42.58%) demonstrated significant and enhanced apoptotic activity in the sub-G1 phase of HeLa cells [191]. Djemoui et al. (2019) synthesized triazole-benzimidazole-based chalcone derivatives as anticancer agents and evaluated them against T47-D, MDA-MB-231, and PC3 cell lines. The cytotoxic effects of triazole-benzimidazole-chalcone hybrid compound **70** were shown at >100, >100, and 5.64 µM, respectively. Compound **70** produced better cytotoxicity in PC3 cell lines. Control, doxorubicin, had IC_50_ values 0.13, 1.51, and 0.73 µM, respectively [192].

Suryanarayana et al. (2021) designed and synthesized novel 1,2,3-triazole-containing dinitrophenylpyrazoles and evaluated them as anticancer agents. These novel molecules were examined for anticancer activity against three tumor cell lines, namely human epithelial colorectal adenocarcinoma (CECA), cervical carcinoma (HeLa), and breast adenocarcinoma (MCF-7). Compound **71** showed potent activity at IC_50_ values 08, 06, and 12 μM, respectively, whereas control combretastatin-A4 had IC_50_ values of 10, 9 and 11 μM, respectively [193]. Chandrashekhar et al. (2016) synthesized triazole- myrrhanone hybrids and tested them for anticancer activity against various cell lines, including A549, Hela, MCF-7, DU-I45, and HepG2. Among them, compound **72** demonstrated good anticancer activity with IC_50_ values of 06.16, 07.76, 09.59, 08.83, and 09.52 μM. Control doxorubicin had IC_50_ values 2.818, 2.570, 1.135, 1.412, and 3.013 µM, respectively [194].

Najafi et al. (2014) synthesized and assessed the in vitro cytotoxic activity of novel triazole- isoxazole derivatives anticancer activity. Most of the synthesized compounds display good anticancer properties. Compound **73** proved effective against MCF-7 and T47D with IC_50_ values of more than 100 and 27.7 μM, respectively, whereas control etoposide had IC_50_ values 7.5 and 7.9 μM, respectively [195]. Duan et al. (2013) synthesized 1,2,3-triazole and dithiocarbamate hybrids and explored their anticancer activity. Most of the synthesized compounds demonstrated good anticancer activity. Among them, compound **74** demonstrated potent activity against four cell lines, i.e., MGC-803, MCF-7, PC-3, and EC-109, with IC_50_ values of 0.73, 5.67, 11.61, and 2.44 μM, respectively. Control, 5-fluorouracil had IC_50_ values of 7.01, 7.54, 27.07, and 3.34 μM, respectively [196]. kumbhare et al. (2015) synthesized and assessed the anticancer activity of triazole thiazole hybrids against various cell lines. Most of the synthesized compounds displayed potent anticancer activity. Compound **75** exhibited potent activity against four cell lines, i.e., MCF-7, A549, A375, and MCF-10A, at IC_50_ values of 2.12, 5.48, 4.7, and 29.33 μM, respectively. Control, doxorubicin, had IC_50_ values 0.12, 3.13, 7.2, and 24.0 μM and paclitaxel had IC_50_ values 2.58, 4.9, 8.0, and 38 µM respectively [197].

Ying et al. (2015) synthesized and reported triazole pyrimidine urea hybrids as anticancer agents. Most of the synthesized compounds displayed potent anticancer activity. Among them, compound **76** displayed potent activity against four cell lines, i.e., EC-109, MCF-7, MGC-803, and B16-F10, with IC_50_ values of 2.96, 3.11, 3.60, and 4.55 µM, respectively. Standard drug, 5-fluorouracil had IC_50_ values 11.61, 9.12, 8.43, and 1.43 µM, respectively [198].

Above reported triazole compounds (Figure 9 (**67**–**76**)) showed the best cytotoxic activity against representative cell lines among their own series. Furthermore, compound **68** (having nitro substituted aromatic ring and imidazole ring joint by ethylene bride with sulphur linkage) showed the lowest IC_50_ value, 0.38 µM, against MCF-7 cell lines as compared to standard drug, doxorubicin, having IC_50_ 0.65 µM.

### 3.8. β-Lactam Derivatives as Anticancer Agents

“β-lactams” represent the most well-known class of antibiotics. Since the 1920s, various new penicillin compounds have been developed, along with related beta-lactam families, including cephalosporins, cephamycins, monobactams, and carbapenems. Every new class of beta-lactam has been created to either deal with particular resistance mechanisms that have emerged in the targeted bacterial population or to broaden the spectrum of activity to encompass more bacterial species. Bacterial enzymes, which hydrolyze the beta-lactam ring and leave the drug inert, represent the main cause of resistance to beta-lactams [199]. Banik et al. (2003) synthesized beta-lactam-based heterocyclic derivatives and evaluated their anti-neoplastic activity on nine representative cell lines. All the synthesized compounds showed good anticancer activity when tested on BRO, MCF-7, MDA-231, OVCAR, SKOV, PC-3, HL-60, K-562, and HT-29 cell lines. Among them, compound **77** showed potent activity at IC_50_ values 10.84, 9.81, 11.98, 4.17, 6.88, 16.32, 3.64, 4.33, and 5.66 µM, respectively. Among the tested cell lines, compound **77** showed excellent activity against HL-60 cell lines with IC_50_ values at 3.64 µM. Cisplatin was used as a reference drug with IC_50_ values 7.66, 10.05, 12.33, 3.99, 5.99, 4.66, 1.66, 2.33, and 16.99 µM, respectively [200].

Borazjani et al. (2020) synthesized beta-lactams-based heterocyclic derivatives and evaluated their anticancer activity on three representative cell lines. All the synthesized compounds showed potent anticancer activity against MCF-7, TC-1, and HepG2 cell lines. Among them, compound **78** showed good anticancer activity at IC_50_ values of 131.52, 85.34, and >1000 µM respectively. Among the tested cell lines, compound **78** showed potent activity against TC-1 cell lines with IC_50_ value 85.34 µM. Gemcitabine was used as a reference drug with IC_50_ values 191.57, 153.25, and 215.01 µM, respectively [201]. Nagaraju Payili et al. (2018) synthesized and evaluated anticancer activity of beta-lactam based derivatives. Utilizing the MTT assay and in vitro antiproliferative activity, newly synthesized molecules were evaluated. Out of the 14 quinone bearing carbamyl-lactam-lactam hybrids synthesized, compound **79** exhibited excellent anticancer activity against the B16F10 cell line. However, the experiments also revealed that the carbamyl derivative with a methyl group at the piperidine rings on the 4-position displayed better activity with IC_50_ value 24.40 μM [202]. Ranjbari S. et al. (2020) synthesized beta-lactam-based heterocyclic derivatives and evaluated their anticancer activity on four representative cell lines. All the synthesized compounds showed potent anticancer activity against SW1116, HepG2, MCF-7, and HEK-293 cell lines. Among them, compound **79** showed potent anticancer activity at IC_50_ values 1.53, ˃ 100, ˃ 100, and ˃ 100 µM respectively. Among the tested cell lines, compound **80** showed most potent activity against SW1116 cell lines with IC_50_ value 1.53 µM. Doxorubicin is used as a reference drug with IC_50_ 6.9, 7.3, 5.5, and 6.1 µM respectively [203].

Fabian E. Olazaran et al. synthesized beta-lactams-based heterocyclic derivatives and evaluated their anticancer activity on three representative cell lines. All the synthesized compounds showed potent anticancer activity when tested on SiHa, B16F10, and CHANG cell lines. Among the synthesized compounds, compound **81** showed potent activity at IC_50_ value 0.07, 1.21, and > 10.0 µM respectively. Among the tested cell lines, compound **81** showed strong activity against SiHa cell lines with IC_50_ value 0.07 µM. Vincristine was used as reference drug with IC_50_ values 0.01, 0.01, and 0.01 µM, respectively. Furthermore, compound **81** showed induction of apoptosis [204]. Rashidi et al. performed MTT assay for newly synthesized compounds in which cell viability was determined at fixed concentrations of 2 and 4 µg/mL, all compounds showed moderate to high cytotoxic activity in MCF-7 cell lines. Among them, compound **82** showed good % cell viability of 21.19 and 27.35% with control 2,4-D (80.40 and 53.10% respectively) [205]. Fu et al. (2017) synthesized beta-lactams-based heterocyclic derivatives and evaluated their anticancer activity on three representative cell lines. All the synthesized compounds showed potent antineoplastic activity when tested on MGC-803, MCF-7, and A549 cell lines. Among them, compound **83** showed most potent activity at IC_50_ values 0.106, 0.421, and 0.507 µM respectively. Among the tested cell lines, compound **83** showed potent activity against MGC-803 cell lines with IC_50_ value 0.106 µM. Fosbretabulin (CA-4P) was used as a control with IC_50_ values 0.015, 0.023, and 0.033 µM respectively [206]. Gupta et al. (2015) designed and synthesized beta-lactam based hetrocyclic derivatives, evaluated their anticancer activity on three representative MCF-7, MDA-MB- 231, and NCI60 cell lines. Among the synthesized compounds, compound **84** showed potent anticancer activity against MCF-7 cell lines with IC_50_ value 7 µM. The control used was Combretastatin A-4 with IC_50_ value 0.015 µM respectively [207].

Banik et al. synthesized beta-lactam-based heterocyclic derivatives and evaluated their anticancer activity on seven representative cell lines. All the synthesized compounds showed potent anticancer activity when tested on BRO, MDA-231, SKOV-3, PC-3, HL-60, K-562, and HT-29 cell lines. Among them, compound **85** showed good anticancer activity at IC_50_ value 6.1, 0.8, 6.8, 1.4, 0.7, 1.1, and 0.7 µM respectively. Among the tested cell lines, compound **85** showed excellent activity against HT-29 cell lines with IC_50_ value 0.7 µM. Cisplatin was used as a control with IC_50_ values 7.66, 12.33, 5.99, 4.66, 1.66, 2.33, and 16.99 µM, respectively [208].

Miriam Carr et al. (2010) designed and synthesized beta-lactams-based heterocyclic derivatives and evaluated their anti-proliferative activity on two representative cell lines. All the synthesized compounds showed good anticancer activity when tested on MCF-7 and MDA-MB-231 cell lines. Among them, compound **86** showed potent activity at IC_50_ values 0.017 and 0.054 µM, respectively. Among the tested cell lines, compound **86** showed the most potent activity against MCF-7 cell lines with IC_50_ value 0.017 µM. 2(CA-4) was used as a control with IC_50_ values 0.0031 and 0.043 µM, respectively. Additionally, the most potent compound **86** exhibited anti-proliferative action at nanomolar doses. Like CA-4, compound **105** inhibited tubulin polymerization, leading to a G2/M phase cell cycle arrest in human MCF-7 breast cancer cells [209]. In summary, the above reported beta-lactam derivatives (Figure 10 (**77**–**86**)) were acting as anticancer agents. The most potent compound among them was compound **86** with 3- hydroxy-4-methoxyphenyl-1-3,4,5-trimethoxyphenyl terminal moiety, having the lowest IC_50_ value of 0.017 µM against MCF-7 (breast cancer) cell line.

### 3.9. Indole Derivatives as Anticancer Ahents

Benzene and pyrrole rings are fused to form the heterocyclic molecule known as indole. A.V. Baeyer and C.A. Knop identified the compound, indole, as the fundamental component of the naturally occurring dye indigo. Indole was detected in coal tar by R. Weissgerber in 1910. It has an m.p. between 52 and 54 °C and a b.p. of 254.7 °C. It is an opaque solid. It is readily soluble in most solvents, including benzene, diethyl ether, and ethanol. Additionally, it is volatile in steam, barely soluble in cold water, and miscible in hot water. Since indole comes from a natural source and is a component of orange blossom oil and jasmine oil, it is found in many perfumes, has been used for a long time to mask unpleasant odors, and has crucial role in the synthesis of the amino acid, tryptophan [210].

Singla et al. (2018) designed and synthesized indole-based derivatives and evaluated their anticancer activity on using T47D cell line. Among them, compound **87** showed good anticancer activity against T47D cell lines with IC_50_ value 3.8 µM, compared to the control drug, bazedoxifene, with IC_50_ value 16.43 µM. These experiments showed that methyl and dimethyl-substituted compounds were more active than aryl-substituted ones. It was confirmed that compound **87** inhibited the expression of the ER-α mRNA in the T47D cells [211]. Cihan-Ustundag et al. (2015) synthesized indole-based derivatives and evaluated their anticancer activity on nine representative cell lines. All the synthesized compounds showed potent anticancer activity when tested on CCRF-CEM, A549/ATCC, COLO 205, SF-268, LOX IMVI, IGROV1, 786-0, PC-3, and MCF7 cell lines. Among them, compound **88** showed the most potent activity at GI_50_ values of 0.033, 0.034, 0.018, 1.32, 0.028, 0.032, 0.031, 0.034, and 0.029 µM, respectively. Among the tested cell lines, compound **88** showed potent activity against COLO 205 (Colon cancer) cell lines with GI_50_ value 0.018 µM. 5-fluorouracil used was a control and had GI_50_ values 10.0, 0.20, 0.16, 1.58, 0.25, 1.26, 0.79, 2.51, and 0.079 µM respectively [212]. He et al. (2019) synthesized indole-based derivatives and evaluated their anticancer activity on five representative cell lines. All the synthesized compounds showed potent anticancer activity against PC3, MGC803, EC109, WPMY-1, and GES-1 cell lines. Among them, compound **89** showed good activity at IC_50_ value 0.14, 2.94, 3.99, 9.85, and 9.23 µM respectively. Among the tested cell lines, compound **89** showed excellent activities against PC3 cell lines with IC_50_ value 0.14 µM. Cisplatin was used as a control and had IC_50_ values 5.44, 1.82, 4.21, 17.25, and 19.02 µM respectively. Furthermore, apoptosis was successfully induced, and PC3 cell proliferation and colonization were effectively inhibited by compound **89**. It did not affect the cell cycle, but it may drastically reduce invasion and migration by inhibiting the EMT process [213]. Prakash et al. (2017) designed and synthesized indole-based derivatives for their potential anticancer activity on the HeLa cell line using MTT assay. The compound **90** showed potent anticancer activity on tested cell lines. Moreover, compound **90** containing an amino group showed potent activity against (HeLa) cervical cancer cell lines with IC_50_ value 13.41 µM, whereas control cisplatin had an IC_50_ value 13.20 µM. The anticancer effectiveness was increased by the amino group in pyrimidine [214].

Imran Ali et al. (2018) designed and synthesized indole based heterocyclic derivatives and evaluated their anticancer activity on HepG2/C3A line. Among them, compound **91** showed anticancer activity at the micellar concentration of 670 µL mL^1^. By using Lipinski’s “rule of five,” drug-likeness properties were predicted. Through interactions and electrostatic attraction, the compound **90** was bound to DNA with a K_b_ value 1.0 × 10^5^ M^−1^. Compound **91** had 1.44% DLC (drug loading content) and 14.4% DLE (drug-loading efficiency) [215].

Chen et al. (2018) designed and synthesized indole-based derivatives and evaluated their anticancer activity on five representative cell lines. All the synthesized compounds showed potent anticancer activity when tested on MCF-7, A549, HepG2, HeLa, and 293T cell lines. Among them, compound **92** showed most potent activity at GI_50_ values 0.09, 0.59, 0.029, 0.034, and >300 µM respectively. Among the tested cell lines, compound **92** showed excellent anticancer activity against MCF-7 cell lines with IC_50_ value 0.09 µM. CA-4 was used as a reference with GI_50_ values 0.14, 0.31, 0.17, 0.092, and >300 µM, respectively [216].

Lafayette et al. (2017) synthesized indole-based derivatives and evaluated their anticancer activity on four representative cell lines. All the synthesized compounds showed potent anticancer activity on HL60, K562, T47D, and MCF7 cell lines. Among them, compound **93** showed good anticancer activity at IC_50_ values 12.78, > 50, 1.93, and > 50 µM respectively. Among the tested cell lines, compound **93** showed most potent activity against T47D cell lines with IC_50_ value 1.93 µM. Doxorrubicin was used as a control and had IC_50_ values 0.01, 6.25, 4.61, and 2.71 µM respectively [217]. Yali Song et al. (2020) designed and synthesized indole-based derivatives and evaluated their anticancer activity on five representative cell lines. All the synthesized compounds showed potent antineoplastic activity on MCF-7, Hela, MGC-803, Bel-7404, and L929 cell lines. Among them, compound **94** showed potent activity at IC_50_ values 25.60, 24.96, 20.54, 21.02, and 105.1 µM respectively. Among the tested cell lines, compound **94** showed most potent activity against Bel-7404 cell lines with the IC_50_ value 21.02 µM. Etoposide was used as a standard with IC_50_ values 1.45, 5.35, 11.18, 7.76, and 80.36 µM, respectively. Furthermore, Compound **113** responded by arresting their cell cycle in the G2/M phase [218]. Zohreh Bakherad et al. (2019) design and synthesized indole-based derivatives and evaluated their anticancer activity on four representative cell lines. All the synthesized compounds showed potent anticancer activity when tested on A-549, Hep-G2, T-47D, and MCF-7 cell lines. Among them, compound **95** showed very good activity at IC_50_ values 25.67, 35, 79, >100, and >100 µM respectively. Among the tested cell lines, compound **95** showed excellent activity against A-549 cell lines with the IC_50_ value 25.67 µM. Etoposide was used as a standard with IC_50_ values 1.76, 5.93, 7.1, and 4.96 µM, respectively [219].

Eldehna et al. (2020) designed and synthesized indole based derivatives and evaluated their antiproliferative activity on two representative cell lines. All the synthesized compounds showed very good anticancer activity against MCF-7, and MDA-MB-231 cell lines. Among them, compound **96** showed the most potent activity at IC_50_ values 0.44 and 1.32 µM respectively. Among the tested cell lines, compound **96** showed excellent anticancer activity against MCF-7 cell lines with the IC_50_ value 0.44 µM. Staurosporine was used as a reference control and had IC_50_ values 6.81 and 10.29 µM, respectively [220]. Reddymasu Sreenivasulu et al. (2020) synthesized indole-based derivatives and evaluated their anticancer activity on four representative cell lines. All the synthesized compounds showed potent anticancer activity against A549, MDA-MB231, MCF-7, and HeLa cell lines. Among them, compound **97** showed good activity at IC_50_ values 3.3, 10.23, 2.6, and 6.34 µM, respectively. Among the tested cell lines, compound **97** showed potent activity against MCF-7 cell lines with the IC_50_ value 2.6 µM. Doxorubicin was used as a control and had IC_50_ values 0.36, 0.47,0.98, and 0.89 µM, respectively [221].

Analyzing the above reported indole-based derivatives (Figure 11 (**87**–**97**)) discussed herewith, all compounds act as anticancer agents. The most potent compound among them was the compound **88,** which was modified to change the indole substituents, the indole-2-carboxamide/carbohydrazide moiety, and substitute the aromatic ring for improved anticancer efficacy against colon cancer cell line (COLO 205) at the lowest GI_50_ value, 0.018 µM.

### 3.10. Pyrazole Derivatives as Anticancer Agents

The chemical formula for pyrazole is C_3_H_4_N_2_ and it has a five-membered ring structure with three C and two nitrogen atoms nearby. It’s conjugate acid’s pKa at 25 °C is 2.49, making it a weak base with a pKb of 11.5 (pKa). Ludwig Knorr introduced the word pyrazole in 1883. They are categorized as alkaloids because of their chemical nature and distinct pharmacological effects. The first naturally occurring pyrazole was 1-pyrazolyl-alanine, discovered in 1959 from watermelon seeds. Pyrazoles are reported to have a variety of biological activities, including neuroprotective, anti-fungal, antitubercular, anti-inflammatory, anti-convulsant, anticancer, anti-viral, ACE inhibitory, antiviral, cholecystokinin-1 receptor antagonist, and estrogen receptor (ER) agonistic activity [222].

Abdelgawad et al. (2018) designed and synthesized pyrazole-based heterocyclic derivatives and evaluated their antineoplastic activity on two representative cell lines. All the synthesized compounds showed very good anticancer activity against HePG2, and MCF-7 cell lines. Among them, compound **98** showed potent anticancer activity with IC_50_ values 18.23 and 5.23 µM respectively. Among the tested cell lines, compound **98** showed excellent activity against MCF-7 cell lines with IC_50_ value 5.23 µM. DOX (doxorubicin) was used as standard and had IC_50_ values 4.50, and 4.70 µM respectively [223]. Nassar et al. (2017) designed and synthesized pyrazole-based heterocyclic derivatives and evaluated their anti-proliferative activity on three representative cell lines. All the synthesized compounds showed potent anticancer activity against HEPG-2, HCT-116, and MCF-7 cell lines. Among them, compound **99** showed very good anticancer activity with IC_50_ values 0.39, 12.96, and 6.97 µM respectively. Among the tested cell lines, compound **99** showed most potent activity against HEPG-2 cell lines with the IC_50_ value 0.39 µM. DOX (Doxorubicin) was used as standard with IC_50_ values 4.00, 3.73, and 2.97 µM, respectively [224]. El-Sayed et al. (2019) designed and synthesized pyrazole-based heterocyclic derivatives and evaluated their antineoplastic activity on two representative cell lines. All the synthesized compounds showed good anticancer activity against HepG2, and HeLa cell lines. Among them, compound **100** showed very good activity with IC_50_ values 5.16, and 4.26 µM respectively. Among the tested cell lines, compound **100** showed most potent activity against HeLa cell lines with IC_50_ value 4.26 µM. DOX (doxorubicin) was used as standard with IC_50_ values 4.50 and 5.57 µM, respectively [225]. Omrana et al. (2019) designed and synthesized pyrazole-based heterocyclic derivatives and evaluated their anti-proliferative activity on two representative cell lines. All the synthesized compounds showed potent anticancer activity against HepG2 and VERO-B cell lines. Among them, compound **101** showed good anticancer activity with IC_50_ values 2.0 and 9.0 µM, respectively. Among the tested cell lines, compound **101** showed most potent activity against HepG2 cell lines with the IC_50_ value 2.0 µM. DOX (doxorubicin) was used as reference standard with IC_50_ values 5.5 and 5 µM, respectively [226].

Zhu et al. (2013) designed and synthesized pyrazole-based heterocyclic derivatives and evaluated their anticancer activity on four representative cell lines. All the synthesized compounds showed good anticancer activity against SGC 7901, A549, Raji, and Hela cell lines. Among them, compound **102** showed most potent activity with IC_50_ values 2.71, 3.18, 1.09, and 13.52 µM respectively. Among the tested cell lines, compound **101** showed excellent activity against Raji cell lines with the IC_50_ value 1.09 µM. Cisplatin was used as control with IC_50_ values 7.56, 17.78, 17.32, and 14.31 µM, respectively [227].

Alam et al. (2016) designed and synthesized pyrazole-based heterocyclic derivatives and evaluated their antineoplastic activity on four representative cell lines. All the synthesized compounds showed very good anticancer activity against Hela, NCI-H460, PC-3, and NIH-3T3 cell lines. Among them, compound **103** showed good activity with IC_50_ values 7.98, 8.41, 9.53, and 87.30 µM respectively. Among the tested cell lines, compound **103** showed the most potent activity against Hela (human cervix) cell lines with the IC_50_ value 7.98 µM. Etoposide was used as standard control with IC_50_ values 11.25, 16.63, 23.80, and 90.53 µM respectively [228].

Harras et al. (2018) designed and synthesized pyrazole-based heterocyclic derivatives and evaluated their anti-proliferative activity on three representative cell lines. All the synthesized compounds showed good anti-proliferative activity against HCT116, UO-31, and HepG2 cell lines. Among them, compound **104** showed most potent activity with IC_50_ values 0.035, 2.24, and 0.028 µM respectively. Among the tested cell lines, compound **104** showed excellent activity against HepG2 cell lines with the IC_50_ value 0.028 µM. Sorafenib was used as reference standard with IC_50_ values 0.66, 7.15, and 0.37 µM, respectively. Additionally, biological analysis of compound **104** found that it induced CDK1 expression in HepG2 cells and during the G2/M phase cell cycle arrest [229]. Liu et al. (2019) designed and synthesized pyrazole-based heterocyclic derivatives and evaluated their anti-proliferative activity on four representative cell lines. All the synthesized compounds showed good anticancer activity against MDA-MB-231, MCF-7, HepG-2, and SMMC-7721 cell lines. Among them, compound **105** showed most potent activity with IC_50_ values 2.41, 2.23, 3.75, and 2.31 µM respectively. Among the tested cell lines, compound **105** showed most potent activity against MCF-7 cell lines with the IC_50_ value 2.23 µM. Doxorubicin was used as standard with IC_50_ values 2.24, 0.34, 2.96, and 0.79 µM, respectively [230]. Afif et al. (2019) designed and synthesized pyrazole-based heterocyclic derivatives and evaluated their anticancer activity on five representative cell lines. All the synthesized compounds showed good anti-proliferative activity against A549, MCF-7, HepG-2, Caco-2, and PC3 cell lines. Among them, compound **106** showed potent activity with IC_50_ values 18.85, 23.43, 23.08, 23.08, and 18.50 µM, respectively. Among the tested cell lines, compound **106** showed the most potent activity against HepG-2 cell lines with the IC_50_ value 23.08 µM. 5-FU was used as standard with IC_50_ values 83.03, 93.79, 96.89, 112.24, and 82.26 µM, respectively [231].

Bakhotmah et al. (2020) designed and synthesized pyrazole-based heterocyclic derivatives and evaluated their anticancer activity on three representative cell lines. All the synthesized compounds showed potent anticancer activity against HCT-116, Hep-G2, and MCF-7 cell lines. Among them, compound **107** showed very good activity with IC_50_ values 4.13, 5.29, and 6.79 µM respectively. Among the tested cell lines, compound **107** showed the most potent activity against HCT-116 cell lines with the IC_50_ value 4.13 µM. Doxorubicin was used as control drug with IC_50_ values 0.493, 0.467, and 0.469 µM, respectively [232].

Analyzing the above reported pyrazole-based heterocyclic derivatives (Figure 12 (**98**–**107**)) discussed herewith, all compounds act as anticancer agents. The most potent compound among them was compound **104** with thioxodihydropyrimidine-4,6(1H,5H)-dione terminal moiety and two substituted aromatic rings, having the lowest IC_50_ value 0.028 µM against HepG2 cell lines. Authors found that these substances downregulated CDK1 expression in HepG2 cells and cell cycle arrest during the G2/M phase.

### 3.11. Quinazoline Derivatives as Anticancer Agents

The parent compound of quinazoline is naphthalene, a mancude organic heterobicyclic compound in which the carbon atoms at positions 1 and 3 have been substituted with nitrogen atoms. It is an azaarene and has an ortho-fused heteroarene. Quinazoline and quinazolin-none scaffolds constitute a sizeable class of physiologically active nitrogen heterocyclic compounds. Numerous anticancer drugs are marketed based on these moieties, including gefitinib, erlotinib, lapatinib, afatinib, and vandetanib [233].

Ghorab et al. (2016) designed and synthesized quinazoline-based heterocyclic derivatives and evaluated their antiproliferative activity on four representative cell lines. All the synthesized compounds showed very good anticancer activity when tested on A549, HeLa, LoVo, and MDA-MB-231 cell lines. Among them, compound **108** showed potent activity with IC_50_ values 77.8, 91.5, 96.5, and 77.9 µM, respectively. Among the tested cell lines, compound **108** showed the most potent activity against A549 cell lines with the IC_50_ value 77.8 µM. Doxorubicin (DOX) was used as control with IC_50_ values 283.5, 120.7, 374.4, and 26.5 µM respectively [234].

Chang et al. (2017) synthesized quinazoline-based heterocyclic derivatives and evaluated their anticancer activity on five representative cell lines. All the synthesized compounds showed potent anticancer activity against HepG2, A549, DU145, MCF-7, and SH-SY5Y cell lines. Among them, compound **109** showed potent anticancer activity at IC_50_ values 4.61, 9.50, 6.79, 9.80, and 7.77 µM, respectively. Among the tested cell lines, compound **109** showed the most potent activity against HepG2 cell lines with the IC_50_ value 4.61 µM. Gefitinib and QWL-138 were used as controls with IC_50_ values 29.79, 12.08, 8.63, 12.05, 18.21, and 3.06, 6.71, 20.04, 10.54, 8.60 µM, respectively. Compound **108** induced cell death primarily through apoptosis via a mitochondrial-dependent route and cell cycle arrest in the S phase [235].

Ahmed et al. (2022) synthesized quinazoline-based heterocyclic derivatives and evaluated their anti-proliferative activity on two representative cell lines. All the synthesized compounds showed potent anticancer activity on MCF-7 and HCT116 cell lines. Among the tested compounds, compound **110** showed very good anticancer activity at IC_50_ values 4.43 and 8.23 µM with control DOX with IC_50_ values 8.09 µM and 11.26 µM, respectively. Further, they acted on the G2/M phase of cell cycle [236]. Heba S.A. ElZahabi et al. (2021) synthesized quinazoline-based heterocyclic derivatives and evaluated their anticancer activity on two representative cell lines. All the synthesized compounds showed potent anticancer activity against MCF-7 and A-549 cell lines. Among them, compound **111** showed the most potent activity at IC_50_ values 0.06 and ≥100 µM respectively. Among the tested cell lines, compound **111** showed excellent activity against MCF-7 cell lines with IC_50_ value 0.06 µM. Doxorubicin and 5-Fluorouracil were used as references with IC_50_ values 0.06, 0.13 and 2.13, 2.36 µM respectively. Compound **111** boosted apoptotic cell death and induced the arrest of the G1 and pre-G1 phases of the cell cycle [237].

Abdelsalam et al. (2019) synthesized quinazoline-based heterocyclic derivatives and evaluated their anticancer activity on five representative cell lines. All the synthesized compounds showed good antineoplastic activity against HepG2, MCF-7, HCT116, and BHK-21 cell lines. Among them, compound **112** showed the most potent activity at IC_50_ values 24.76, 7.70, 23.66, and >200 µM respectively. Among the tested cell lines, compound **112** showed excellent activity against MCF-7 cell lines with the IC_50_ value 7.70 µM. Dox and Erlotinib were used as standards with IC_50_ values 1.95, 1.10, 0.63, and 3.04 and 10.19, 5.06, 13.22, and 19.13 µM respectively [238].

Hoan et al. (2019) synthesized quinazoline-based heterocyclic derivatives and evaluated their anticancer activity on four representative cell lines. All the synthesized compounds showed good anti-proliferative activity against KB, Hep-G2, Lu, and MCF-7 cell lines. Among them, compound **113** showed potent activity at IC_50_ values > 337, 2.1, 11.6, and 2.2 µM respectively. Among the tested cell lines, compound **113** showed the most potent anticancer activity against Hep-G2 cell lines with IC_50_ value 2.1 µM. Ellipticine was used as standard with IC_50_ values 1.14, 1.14, 1.71, and 1.70 µM, respectively [239]. Wang et al. (2021) designed and synthesized quinazoline-based heterocyclic derivatives and evaluated their anticancer activity on four representative cell lines. All the synthesized compounds showed potent anticancer activity against H1975, PC-3, MDA-MB231, and MGC-803 cell lines. Among them, compound **114** showed good anticancer activity at IC_50_ values 2.03, 5.37, 8.16, and 6.25 µM respectively. Among the tested cell lines, compound **114** showed most potent activity against H1975 cell lines with the IC_50_ value 2.03 µM. Gefitinib was used as standard with IC_50_ values 9.20, 8.92, 7.12, and 8.19 µM, respectively. Furthermore, compound **114** showed G1 phase mediated cell apoptosis in H1975 cell cycle, which promoted the accumulation of ROS [240].

Zayed et al. (2018) synthesized quinazoline-based heterocyclic derivatives and evaluated their antineoplastic activity on two representative cell lines. All the synthesized compounds showed good anti-proliferative activity against MCF-7 and MDA-MBA-231 cell lines. Among them, compound **115** showed the most potent activity at IC_50_ value 12.44, and 0.43 µM respectively. Among the tested cell lines, compound **115** showed excellent anticancer activity against MDA-MBA-231 cell lines with the IC_50_ value 0.43 µM. Erlotinib was used as standard with IC_50_ 1.14, 2.55 µM respectively [241]. Abuelizz et al. (2017) synthesized quinazoline-based heterocyclic derivatives and evaluated their antineoplastic activity on two representative cell lines. All the synthesized compounds showed good anti-proliferative activity and were tested on HeLa and MDA-MBA-231 cell lines. Among them, compound **116** showed most potent activity at IC_50_ values 1.85 and 2.33 µM, respectively. Among the tested cell lines, compound **116** showed excellent anticancer activity against HeLa cell lines with the IC_50_ value 1.85 µM. Gefitinib was used as standard with IC_50_ values 4.3 and 28.33 µM, respectively [242].

Poudapally et al. (2017) synthesized quinazoline-based heterocyclic derivatives and evaluated their anticancer activity on six representative cell lines. All the synthesized compounds showed good anti-proliferative activity against SKOV3, DU145, THP1, U937, COLO205, and FHC cell lines. Among them, compound **117** showed good activity at IC_50_ values 9.23, 12.33, 8.17, 6.06, 19.04, and 73.0 µM respectively. Among the tested cell lines, compound **117** showed the most potent activity against U937 cell lines with the IC_50_ value 6.06 µM. Etoposide was used as standard and had IC_50_ values 3.84, 6.17, 4.42, 2.89, 5.12, and 49.31 µM respectively [243].

Ewes et al. (2020) synthesized quinazoline-based heterocyclic derivatives and evaluated their anticancer activity on five representative cell lines. All the synthesized compounds showed good antineoplastic activity against HePG2, MCF-7, PC3, HCT-116, and Hela cell lines. Among them, compound **118** showed good antineoplastic activity at IC_50_ values of 10.30, 8.90, 10.54, 11.47, and 13.41 µM, respectively. Among the tested cell lines, compound **118** showed potent activity against MCF-7 cell lines with the IC_50_ value 8.90 µM. Doxorubicin was used as standard with IC_50_ values 4.50, 4.17, 8.87, 5.23, and 5.57 µM, respectively [244].

Analyzing the above reported quinazoline-based heterocyclic derivatives (Figure 13 (**108**–**118**)) discussed herewith, all compounds act as anticancer agents. The most potent compound among them was compound **111** with 3-amino-6,8-dibromo-2-methylquinazolin-4(3H)-one terminal moiety, having the lowest IC_50_ value of 0.06 µM against MCF-7 cell lines.

### 3.12. Quinoxaline Derivatives as Anticancer Agents

The nitrogen-containing heterocyclic molecule quinoxalines, commonly known as benzopyrazines, has a ring complex composed of a pyrazine ring and a benzene ring [245]. The majority of quinoxaline derivatives are produced synthetically. In quinoxaline, two benzene rings combine, expanding the variety of resonance structures available to these systems. It possesses a zero-dipole moment [246]. Due to the wide range of pharmacological activities that quinoxalines exhibit, they have received a lot of interest. It is regarded as a crucial moiety for anticancer medications. Since the 1980s, when certain quinoxalinone derivatives were produced and investigated for their cytotoxic activity, the screening of quinoxaline as an anticancer scaffold has continued [247]. Ismail et al. (2010) synthesized a set of quinoxaline 1,4-di-oxides which were tested for their anticancer activities against the U251 (brain tumors) and Hepg2 (liver carcinoma). Among the synthesized compounds, **119** and **120** showed maximum activity against Hepg2 cell lines with IC_50_ values of 0.77 and 0.50 µg/mL, respectively. Tirapazamine was used as standard and its IC_50_ value against HepG2 was 56.6 µg/mL. For carbamate and acid azide derivatives to be cytotoxic, the 7- methoxy (electron-donating group) was important. The activity was decreased when quinoxaline 1,4-dioxide was converted to quinoxaline 4-oxide [248].

Abbas et al. (2015) synthesized novel quinoxaline derivatives and evaluated their cytotoxicity on NCI-H460, SF-268, and MCF-7 cell lines. Out of the synthesized derivatives, compounds (IC_50_ value) **121** (0.06), **122** (0.01), **123** (0.02), and **124** (0.08 µg/mL) had maximum cytotoxic activity against MCF-7 cell lines. Doxorubicin was used as a standard drug with IC_50_ value for NCI-H460 (0.09), SF-268(0.09), and MCF-7(0.04 µg/mL) cells. Compound **121** had enhanced activity because of the presence of the methoxy. The substitution of an *O*-phenyl linker or an N-phenyl linker in compounds **123** and **124** showed a strong binding affinity and potency [249].

Desplat et al. (2016) prepared novel ethyl pyrrolo quinoxaline-carboxylate derivatives and tested their anticancer activity on five leukemia cells, i.e., Jurkat, K562, U937, HL60, and U266. Compounds A6730 and LY-294002 were used as standard. The standard drug A6730 with IC_50_ of 8 µM and two derivatives **125** (IC_50_ value of 4 µM) and **126** (IC_50_ value of 3 µM) showed excellent activity against U937 cell line. Substitution at the 4th position by a benzylpiperidinyl fluorobenzimidazole group and in the 1st position by a phenyl group produced an active analogue [250]. Newahie et al. (2019) designed and synthesized three series of quinazoline derivatives and evaluated their cytotoxicity against panel of cell lines, including HepG2, HCT116, and MCF-7 cells. From the second series, compound **127** had IC_50_ values of 25.7, 7.8, and 60.3 µM. From third series, compound **128** had IC_50_ values of 22, 2.5, and 9 µM. From the first series, compound **129** had IC_50_ values of 10.0, 4.4, and 5.3 µM, respectively. Doxorubicin was used as standard with IC_50_ value of 1.2, 0.62, and 0.9 µM for HepG2, HCT116, and MCF-7 cells respectively. They also tested some of the compounds for WI-38 cell lines and found that **129** was the most active with greater selectivity for WI-38 cell lines. Compound **129** had a phenyl ring attached to another ring which enhanced activity [251]. Ahmed et al. (2020) prepared new quinoxaline derivatives and tested their cytotoxicity on HepG2, HCT-116, and MCF-7 cells by MTT bioassay using erlotinib and doxorubicin as reference drugs. For given cell lines, compounds **130**, (IC_50_ = 2.51, 4.22 and 2.27 µM), **131** (IC_50_ = 1.32, 1.41 and 1.18 µM), and **132** (IC_50_ = 1.72, 1.85 and 1.92 µM) had maximum cytotoxicity with control erlotinib with IC_50_ values of 1.63, 1.57, and 1.49 µM and doxorubicin with IC_50_ value of 1.41, 0.90, and 1.01 µM, respectively [252].

Alsaif et al. (2021) synthesized new triazolo quinoxaline derivatives and tested them for anticancer activity against HepG2 and MCF-7 cell lines. Standard drug sorafenib had IC_50_ values against HepG2 (3.51μM) and MCF-7 (2.17μM). Some of the triazolo quinoxaline derivatives, i.e., **133, 134, 135,** and **136**, presented comparable activity with reference drug sorafenib against HepG2 cell lines with IC_50_ values of 4.1, 8.4, 7.8, and 9.7 μM, respectively. The capability of the synthesized substances to inhibit VEGFR-2 was further evaluated. Compound **133** was the most potent VEGFR-2 inhibitor, with the IC_50_ value of 3.4 nM, which is close to sorafenib (standard drug) with the IC_50_ value of 3.2 nM [253]. Khaled El-Adl et al. (2021) prepared new quinoxaline-one derivatives and tested their anticancer activity against HCT-116, MCF-7, and HepG-2 cell lines. Doxorubicin and sorafenib were used as positive controls. Against synthesized compounds, HCT-116 was the most sensitive cell line. The highest level of cytotoxicity was displayed by compounds **137** and **139** toward the examined cell lines. Compound **138** was the most active derivative with IC_50_ values against HepG-2 (5.30 μM), MCF-7 (2.20 μM), and HCT-116 (5.50 μM). In comparison to doxorubicin (IC_50_ value of 8.28, 7.67 and 9.63 μM respectively) compounds **138** and **140** showed better anticancer effects against tested cells. In comparison to the hydrophobic and electron-donating groups of other derivatives, the distant benzyl moiety in compound **138** demonstrated better activity [254]. Hajri et al. (2016) synthesized quinoxaline-2-carboxylate and quinoxalin-one derivatives and evaluated their anticancer activity against U87-MG and A549 cell lines. The most cytotoxic arylethynyl derivatives, **141** and **142**, both had the IC_50_ value of 3.3 µM. The antiproliferative effect of quinoxaline was due to the presence of the ethynyl group at third position in quinoxaline molecule. The compounds **141**, **142**, and **143**, with no modification on the aromatic group, addition of a methyl group in the 4th position, or a fluorine atom in the 3rd position, showed maximum activity against both U87-MG and A549 cell lines. Compound **143**, containing a fluorine atom in the 3rd position, exhibited a good anticancer activity against the A549 and U87-MG cells [255]. Analyzing the above reported quinoxaline-based heterocyclic derivatives (Figure 14 (**119**–**143**)), all compounds act as anticancer agents. The most potent compound among them was compound **122** in which the quinoxaline ring was substituted at N-1 atom by acetohydrazide moiety and bromine, having the IC_50_ value of 0.01 µg/mL against MCF-7 cell line.

### 3.13. Isatin Derivatives as Anticancer Agents

Isatin is an endogenous substance found in animals that has a number of pharmacological features, such as anticancer action [256]. In the present era of medicine, there are many challenges that affect human health, especially when it comes to tumors. As a result, new treatments that target tumor cells specifically will inevitably need to be introduced to the armory for treating these malignancies [257]. Aziz et al. (2017) prepared a few sets of isatin-based benzoazine heterocyclic rings, such as isatin- quinoxaline, quinazoline, and phthalazines hybrids. They used three human cancer cell lines, i.e., HT-29, ZR-75, and A-549, for testing the cytotoxicity of all the synthesized compounds. A majority of the synthesized molecules showed potent cytotoxic activity. Sunitinib was used as standard drug with an average IC_50_ value of 8.11 µM. Among synthesized compounds, **144** showed maximum anticancer activity with average IC_50_ value of 5.53 µM [258].

Meleddu et al. (2018) produced isatin-dihydropyrazole derivatives and tested their ability to prevent the growth of tumor cells in different types of human cancer cell lines, namely IGR39 (melanoma), lung carcinoma (A549), glioblastoma (U87), breast adenocarcinoma (MCF-7), invasive ductal carcinoma (BT474), non-small cell lung carcinoma (H1299), BxPC-3 (pancreatic adenocarcinoma) ovarian cancer (SKOV-3), and fibroblast. Among the synthesized compounds, **145** showed maximum activity with EC_50_ values of 0.18, 0.14, 0.23, 0.31, and 0.10 µM against A549, IGR39, U87, MCF-7, and BxPC-3 cell lines. Sunitinib was used as the positive control and was having EC_50_ values of 0.90, 0.96, 2.5, 1.36, 1.54, and 0.30 µM for BT474, MCF-7, BxPC-3, SKOV-3, and H1299 fibroblast, respectively [259]. Eldehna et al. (2015) synthesized isatin-pyridine derivatives and evaluated their anti-proliferative effect on three human tumor cancer cell lines: lung cancer (A549), hepatocellular carcinoma (HepG2), and breast cancer (MCF-7). A majority of the synthesized compounds showed good antiproliferative activity. The synthesized compound **146** showed maximum activity with IC_50_ values of 10.8, 6.3, and 8.7 µM, respectively. Doxorubicin was used as a reference drug having IC_50_ values of 7.6, 6.1, and 6.9 µM for A549, MCF-7, and HepG2 [260]. Wabli et al. (2020) synthesized isatin-indole derivatives and evaluated their anticancer efficacy against HT-29 (colon), ZR-75 (human breast), and A-549 (lung) cell lines. A majority of the prepared compounds show potent cytotoxicity. The most active compound was **147** with IC_50_ values of 0.74, 0.76, and 2.02 µM against ZR-75, A-549, and HT-29 respectively. Sunitinib was used as standard drug with IC_50_ values of 10.14, 8.31, and 5.87 µM for HT-29, ZR-75, and A-549, respectively [261]. Panga et al. (2016) synthesized isatin-benzoic acid derivatives and evaluated them in vitro for anticancer activity against HeLa and MCF-7 cell lines. The synthesized compounds showed good anticancer activity towards both the cells. Compound **148** showed the highest activity towards both cell lines having IC_50_ values of 9.28 and 14.89 µM and Vinblastine was used as positive control here with IC_50_ values of 4.02 and 7.14 µM, respectively [262]. Eldehna et al. (2021) synthesized isatin-thiazolo benzimidazole derivatives and evaluated their cytotoxic activity against MDA-MB-231 and MCF-7 breast cancer cell lines by using sulforhodamine B colorimetric (SRB) assay. The compound **149** shows good anticancer activity with IC_50_ values of 6.50 and 2.02 µM against MDA-MB-231 and MCF-7 cell lines, respectively. Compound **150** with IC_50_ values of 2.60 and 3.01 µM respectively shows better anticancer activity than standard drug staurosporine, having IC_50_ values of 4.29 and 3.81 µM, respectively [257].

All the above reported isatin-based heterocyclic derivatives (Figure 15 (**144**–**150**)) were observed to act as anticancer agents. The most potent compound among them was compound **145** with the EC_50_ value of 0.1 µM against the BxPC-3 cell line, which contained diazole with 2-naphthyl group and 4- CH_3_ substituent ring for optimal antiproliferative activity.

### 3.14. Pyrrolo-Benzodiazepines Derivatives as Anticancer Agents

Many actinomycetes species naturally produce or contain pyrrolo-benzodiazepines (PBD) derivatives. PBD binds covalently to DNA, inhibiting transcription factors and DNA replication, thus slowing cellular growth [263]. Bose et al. (2012) synthesized pyrrole-benzodiazepine derivatives and evaluated the cytotoxic activities of the synthesized compounds against HL-60 (human promyelocytic leukemia), THP-1 (human acute monocytic leukemia), U-937 (human histiocytic lymphoma), A-549 (lung carcinoma), and Jurkat (Human T-cell leukemia) cell lines. The most active compound was **151** with IC_50_ values of 0.49, 4.13, 3.44, and 6.58 µM against THP-1, HL-60, U-937, and Jurkat cell lines. THP-1, U-937, HL-60, and Jurkat leukemia cell lines were all cultivated in RPMI-1640, while A-549 was grown in Dulbecco’s Modified Eagle Medium (DMEM). Etoposide used as reference drug had IC_50_ values 2.16, 17.94, 1.83, and 5.35 µM against THP-1, U-937, HL-60, and Jurkat cells [264].

Kamal et al. (2012) synthesized pyrrolo-benzodiazepine conjugated with benzo-indolone derivatives. The synthesized compounds were evaluated for antiproliferative activity in human cancer cell lines of the prostrate, skin, lung, colon, and by using the MTT assay. A majority of the synthesized derivatives showed good anticancer activity. Compound **152** showed maximum activity with IC_50_ values of 1.21, 1.72, 1.05, and 1.52 µM against Colo-205, A431, A549, and PC-3, respectively. Doxorubicin was used as the positive control and had IC_50_ values of 1.69, 0.03, 1.02, and 2.51 µM [265]. Chen et al. (2013) synthesized the new series of pyrrolo-benzodiazepine-triazole derivatives and their anticancer activity was evaluated on A375 cells. Most of the synthesized derivatives showed potent cytotoxic activity. On A375 cells, compound **153** showed a greater inhibitory effect with the IC_50_ value of 2.2 µM [266]. All the above reported Pyrrolo-benzodiazepines-based heterocyclic derivatives (Figure 16 (**151**–**153**)) discussed herewith were observed to act as anticancer agents. The most potent compound among them was compound **151** with the IC_50_ value of 0.49 µM against THP-1, containing a fluorine analogue which may have a role in cell-permeability, which was generally enhanced by fluoro compounds.

### 3.15. Pyrido[2,3-d] Pyrimidine Derivatives as Anticancer Agents

Purines, quinazolines, pteridines, and pyrido-pyrimidines are examples of bicyclic nitrogen-containing heterocyclic compounds that are well-known pharmacophores in medicinal chemistry. Examples of commercial medications with a bicyclic main structure include the tyrosine kinase inhibitors gefitinib and erlotinib, and both are quinazoline derivatives. Both of them are used to manage non-small cell lung cancer. Pyrido[2,3-d]pyrimidines have been studied extensively as quinazoline analogs [267] Pyrido[2,3-d]pyrimidines have shown antitumor, antibacterial, CNS depressive, anticonvulsant, antipyretic, and analgesic effects [268]. Kumar et al. synthesized isoxazole/triazole attached 7(trifluoromethyl) pyrido[2,3d] pyrimidine compounds and evaluated their cytotoxicity against PANC1, HeLa, A549, and MDA MB-231 cells. A majority of the synthesized derivatives showed potent anticancer activity. Compounds **154** and **155** showed good antiproliferative activity against A549 and PANC-1 cell lines. Compound **154** had GI_50_ values of 0.86 and 0.02 µM against A549 and PANC-1 cell lines, respectively, and compound **155** had the GI_50_ value of 0.03 and 0.73 µM against A549 and PANC-1 cell lines, respectively. Nocodazole was used as a standard drug with IC_50_ values 0.08 and 0.029 µM, respectively [269].

Hou et al. (2016) prepared a series of new pyrido[2,3-d] pyrimidine derivatives and tested their cytotoxic activity on five human cancer cell lines: BT-474, SK-BR-3, A549, MCF-7, and MDA-MB-231. Most of the prepared compounds showed potent anticancer activity. Compound **156** showed good activity with IC_50_ values of 1.63, 17.17, 29.73, 13.47, and 7.11 µM against MCF-7, SK-BR-3, A549, BT-474, and MDA-MB-231 cell lines which were comparable to standard gefitinib with IC_50_ values of 25.37, 4.92, 24.25, 0.53, and 37.82 µM, respectively [270].

Banda et al. (2018) synthesized the pyridopyrimidine derivatives and tested them for anticancer activities in four types of human cancer cell lines. The anticancer activity was tested against Colo205, B16F10, U937, and THP-1. Most of the derivatives showed cytotoxic activities against all cells at concentrations lower than 100 µg/mL. Compound **157** was found to be the most effective of these derivatives since it had an IC_50_ value of 19 µg/mL against B16F10 cell line. Further, 5-Fluorouracil was used as a reference drug which showed IC_50_ value of 4.03, 9.82, 0.87 and 0.54 µg/mL against Colo205, B16F10, U937 and THP-1 cell lines respectively [271]. Behalo et al. (2017) produced new series of pyrido[2,3-d] pyrimidine derivatives and tested their anticancer activity against two cell lines: PC-3 and MCF-7. Most of the synthesized derivatives displayed potent anticancer activity. It was found that compounds **158** and **159** had the strongest cytotoxic effect against the PC-3 cell lines with IC_50_ values of 9.47 and 10.34, respectively. 5-Flurouracil was used as a standard drug, having IC_50_ values of 4.91 and 4.73 µg/mL for PC-3 and MCF-7 [272]. Elzahabi et al. (2018) synthesized substituted pyrido[2,3-d] pyrimidines and evaluated their antiproliferative activity against five tumor cell lines, PC-3, HepG-2, MCF-7, A549, and HCT-116. Most of the compounds displayed good anticancer activity. Against HCT-116, HepG-2, PC-3, and A549, compound **160** demonstrated strong anticancer activity with IC_50_ values of 7, 0.3, 6.6, and 9.6 µM, respectively, and reference drug doxorubicin had IC_50_ values of 12.8, 0.6, 6.8, and 0.087 µM, respectively [273].

Faidallah et al. (2016) prepared pyrido[2,3-*d*] pyrimidines and evaluated their cytotoxic activity against three human cancer cell lines HepG2, MCF-7, and HT29. Most of the synthesized derivatives showed potent anticancer activity. Many of the compounds showed good anticancer activity. Compounds **161** (with LD_50_ = 64.6, 6.4 and 25.2 µM), **162** (with LD_50_ = 70.1, 7.9 and 28.8 µM)**,** and **163** (with LD_50_ = 71.2, 8.91 and 26.9 µM) presented comparable anticancer activity to that of doxorubicin with the LD_50_ value of 3, 4, and 40 µM [274]. Warhi et al. (2022) prepared pyrido[2,3-d] pyrimidines derivatives and assessed their anticancer activity against Hepg2 and MCF-7 cell lines. The majority of compounds showed potent anticancer activity. Compound **164** demonstrated comparable activity against the MCF-7 and HepG2 cell lines with IC_50_ values of 6.22 and 19.58 µM, respectively, while the control Taxol had IC_50_ values 8.48 µM and 14.60 µM, respectively [275].

Ding et al. (2022) prepared a series of DHA (dihydroartemisinin) and ARS (artesunate) containing pyrrolo(pyrido)[2,3-*d*] pyrimidine derivatives and tested their antiproliferative activity against MDA-MB-436 and T-47D cell lines. Many of the synthesized compounds showed potent anticancer activity. Among synthesized compounds, **165** displayed maximum anticancer activity towards MDA-MB-436 and T-47D with IC_50_ values of 10.52 and 3.02 µM. Ribociclib was used as positive control with IC_50_ values 24.35 and 4.81 µM, respectively [276].

All the Pyrido[2,3-d]pyrimidine based heterocyclic derivatives (Figure 17 (**154**–**165**)) discussed herewith were observed to act as anticancer agents. The most potent compound among them is compound **163** with GI_50_ of 0.02 µM against PANC-1, due to the presence of an ethyl group at the C-2 position and substituted aromatic ring with fluorine group.

## 4. Conclusions

Heterocyclic systems have earned their place as true pillars of medicinal chemistry owing to their innate ingenuity, adaptability, and outstanding physicochemical potencies. The majority of natural products and pharmaceuticals that are currently prescribed contain the primary heterocyclic systems. Nitrogen heterocycles stand out among them because about 60% of the FDA-approved pharmaceuticals are nitrogen-based heterocycles. The current study deduced that chemically synthesized nitrogen-containing heterocycles (pyrimidine, quinolone, carbazole, pyridine, imidazole, benzimidazole, triazole, β-lactam, indole, pyrazole, quinazoline, quinoxaline, isatin, pyrrolo-benzodiazepines, and pyrido[2,3-d]pyrimidines) are important and versatile molecules against the different forms of cancer. The spectrum of nitrogen-based molecules used in medicine is expanding every day and their numerous analogues offer a promising and significant route for the discovery of medications with a variety of biological applications.

In vitro studies compiled herewith revealed that imidazole, pyridine, quinoxaline, β-lactam, quinazoline, and isatin majorly act against breast cancer (MCF-7), pyrimidine and pyrazole derivatives against liver cancer (HepG2), quinoline and triazole derivatives against cervical cancer (HeLa), carbazole, indole, and pyrrolo-benzodiazepines derivatives against lung cancer (A549), and bezimidazole and pyrido[2,3-d]pyrimidine against colorectal cancer (HCT-116).

Particularly, pyridine derivative compound **39a** showed excellent anticancer activity on all tested cell lines, including MDA-MB-23, A549, and HeLa cell lines at IC_50_ values of 0.0031, 0.089, and 0.0038 µM, respectively. Further, Compounds **10**, **13**, **25a**, **49**, **59**, **68**, **86**, **88**, **104**, **111**, **122**, **145**, **151**, and **164** also showed potent anticancer activity on MCF-7, HeLa, HepG2, EGFR, VEGFR-2, MCF-7, MCF-7 Colo-205, HepG2, MCF-7, MCF-7 BxPc3, THP-1, and PANC-1 at concentration of 0.23, 0.8, 0.012, 0.47, 0.02, 0.38, 0.017, 0.018, 0.028, 0.06, 0.01, 0.1, 0.49, and 0.02 µM to nM range respectively. Among these potent compounds, compounds **10**, **68**, **86**, **111**, and **122** showed excellent activity against the breast cancer (MCF-7 cell line).

The compiled studies confirm the potential of nitrogen heterocyclic molecules in cancer treatment. The reported promising compounds from every scaffold present excellent activity against the different cell lines and kinase inhibitory assay which could be used in the structural based design of efficacious nitrogen-based anticancer drugs.

## Figures and Tables

**Figure 1 pharmaceuticals-16-00299-f001:**
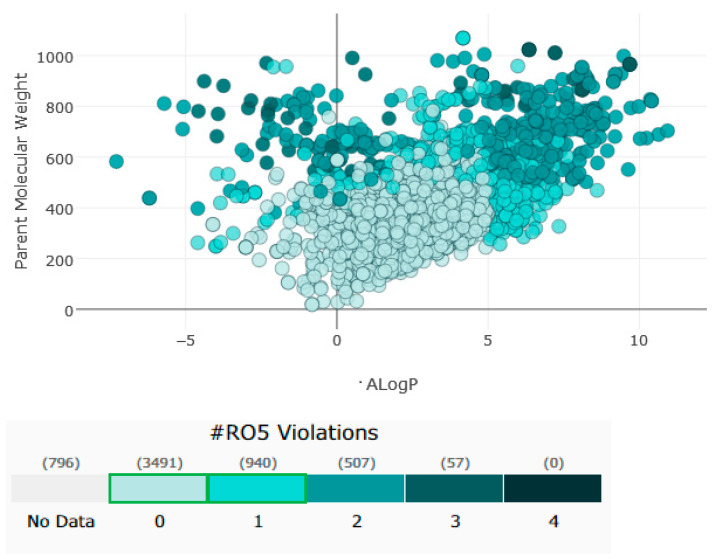
Correlation between molecular weight of nitrogen containing heterocyclic compounds with lop P.

**Figure 2 pharmaceuticals-16-00299-f002:**
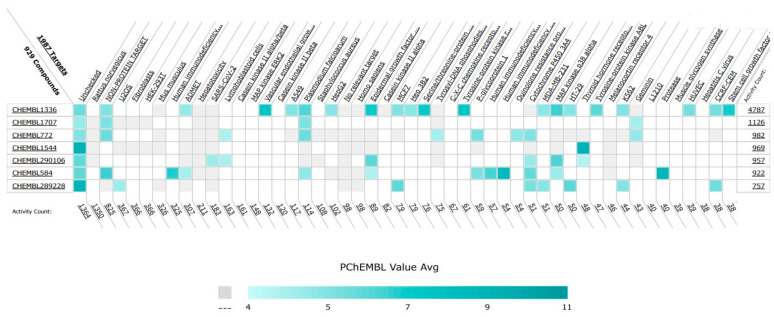
Correlation of nitrogen containing heterocyclic compounds violating RO5 with biological activities.

**Figure 3 pharmaceuticals-16-00299-f003:**
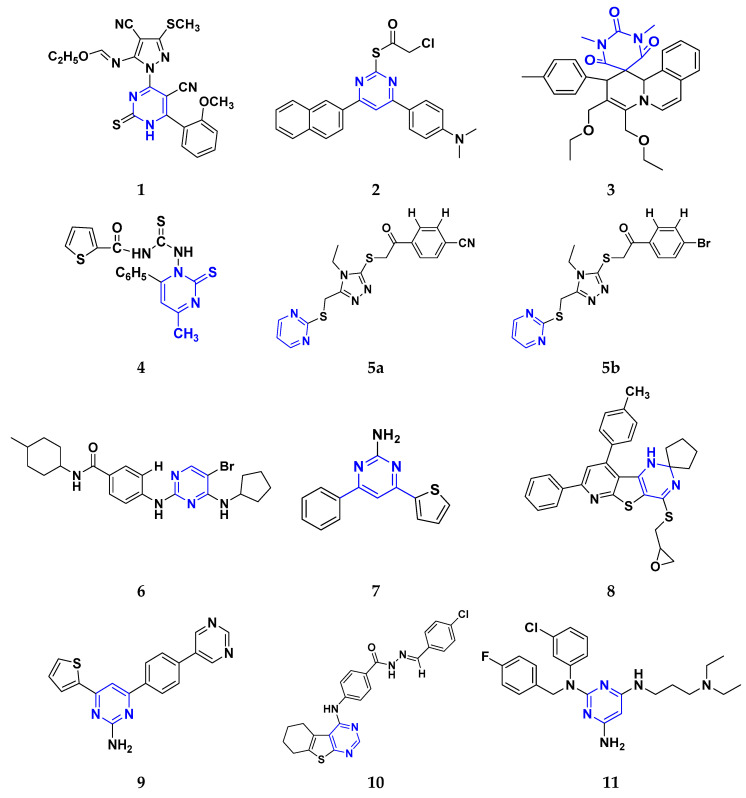
Pyrimidine derivatives (**1**–**11**) as anticancer agents.

**Figure 4 pharmaceuticals-16-00299-f004:**
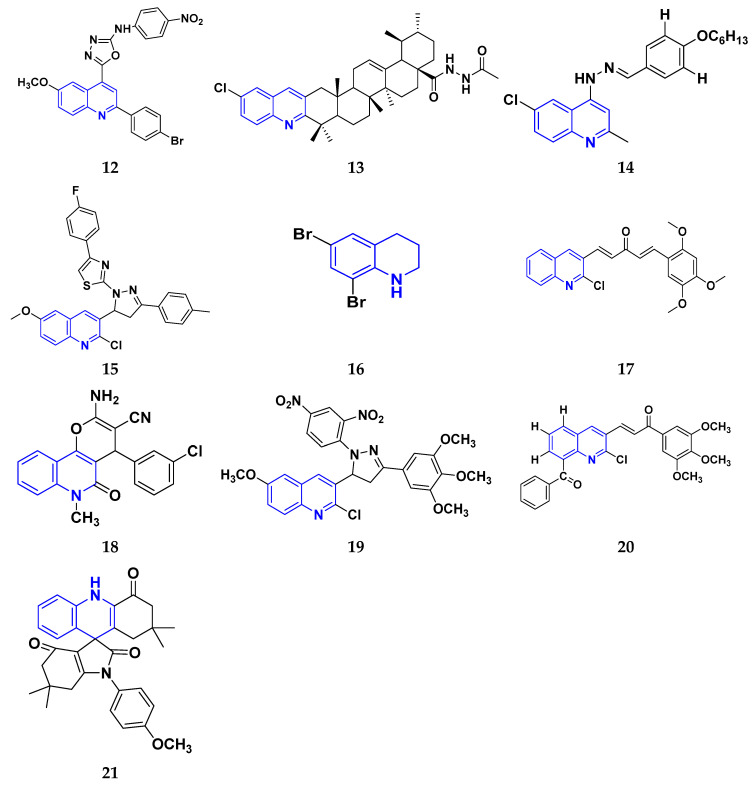
Quinoline derivatives (**12**–**21**) as anticancer agents.

**Figure 5 pharmaceuticals-16-00299-f005:**
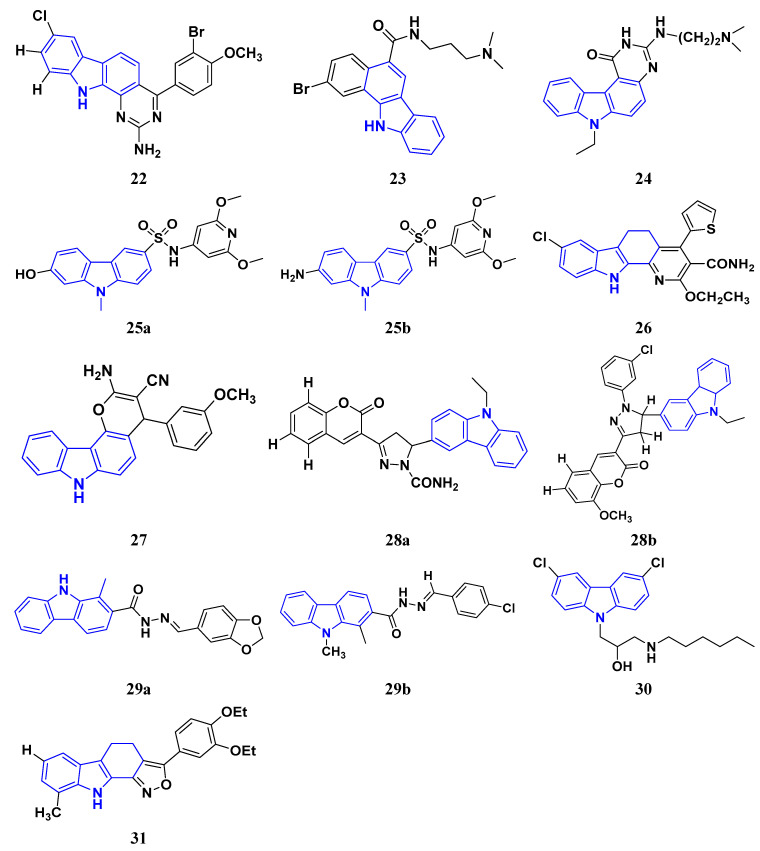
Carbazole derivatives (**22**–**31**) as anticancer agents.

**Figure 6 pharmaceuticals-16-00299-f006:**
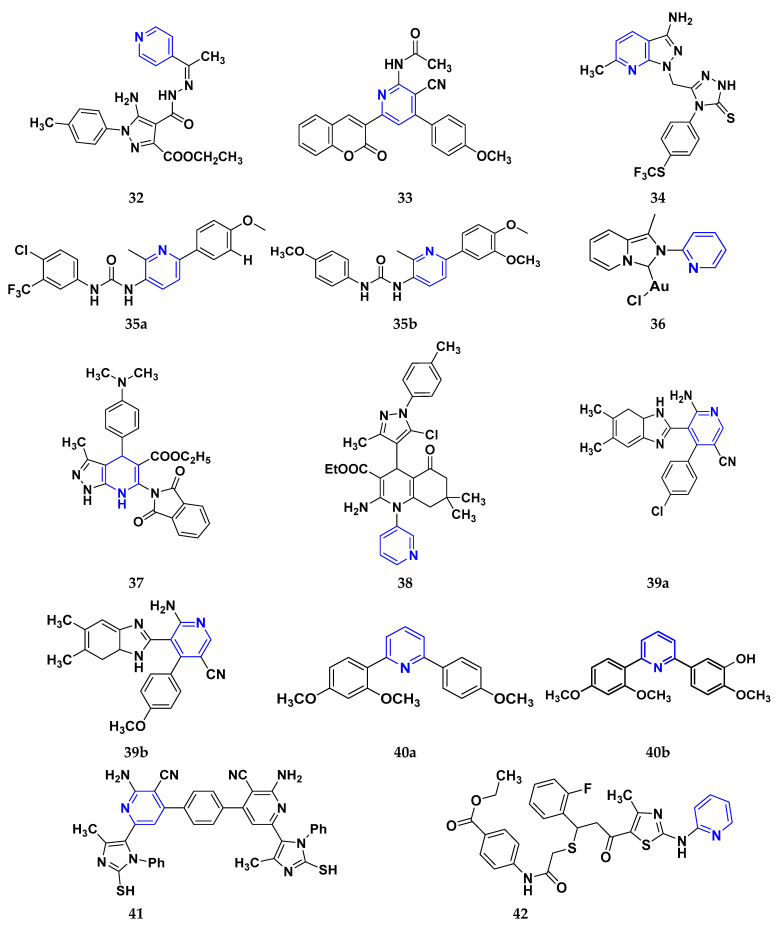
Pyridine derivatives (**32**–**42**) as anticancer agents.

**Figure 7 pharmaceuticals-16-00299-f007:**
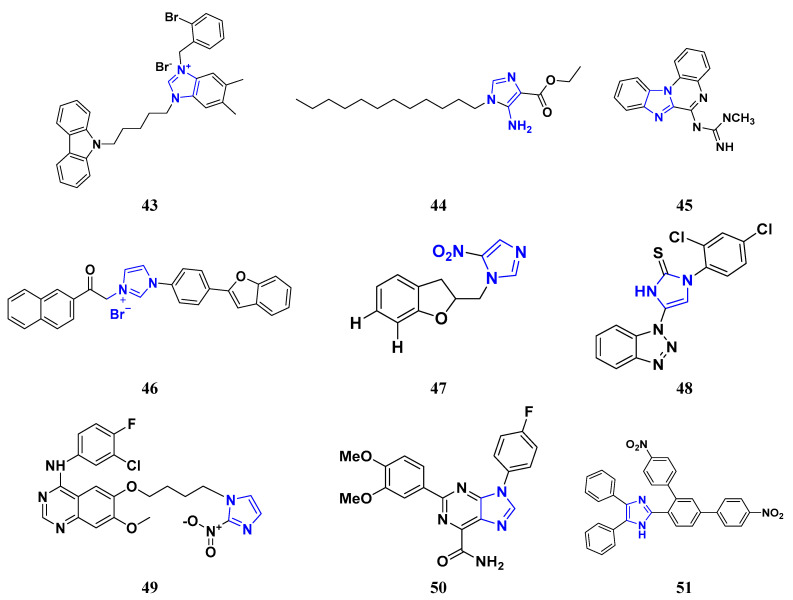
Imidazole derivatives (**43**–**54**) as anticancer agents.

**Figure 8 pharmaceuticals-16-00299-f008:**
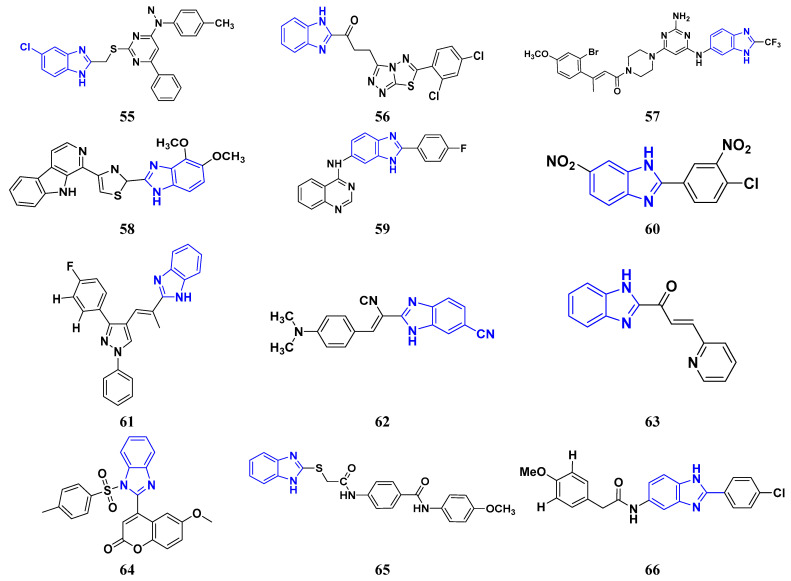
Benzimidazole derivatives (**55**–**66**) as anticancer agents.

**Figure 9 pharmaceuticals-16-00299-f009:**
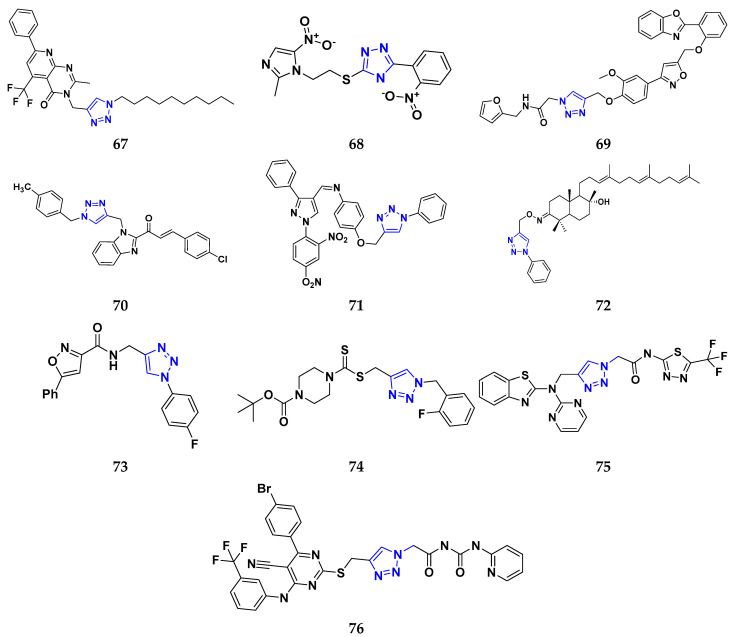
Triazole derivatives (**67**–**76**) as anticancer agents.

**Figure 10 pharmaceuticals-16-00299-f010:**
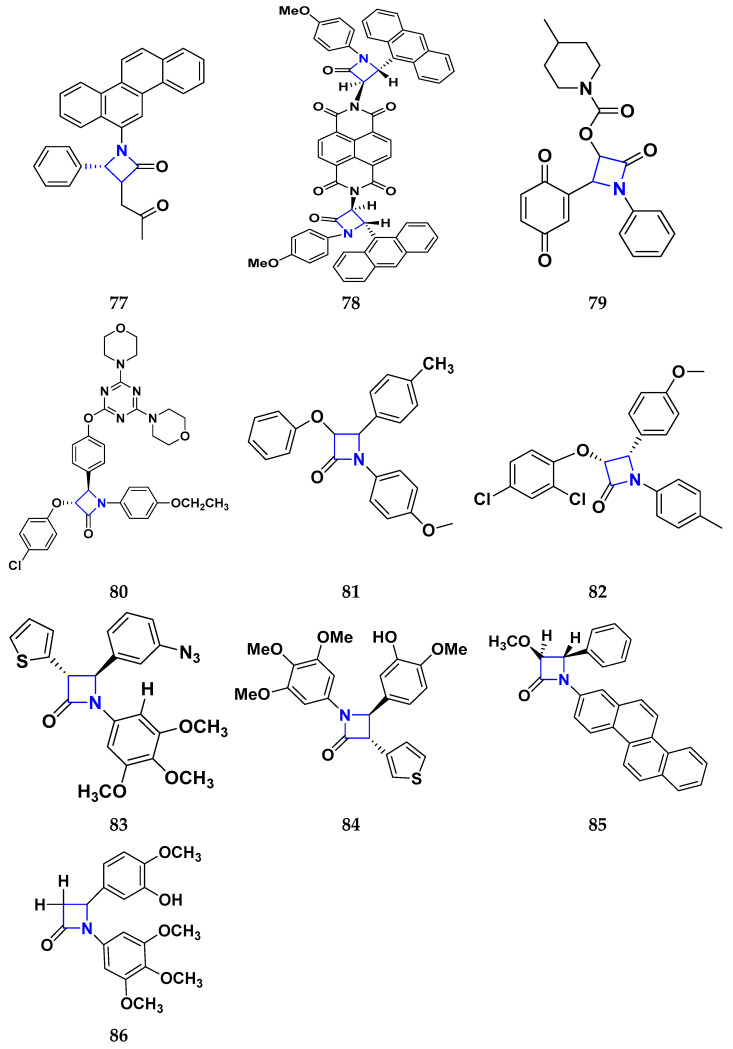
β-lactam derivatives (**77**–**86**) as anticancer agents.

**Figure 11 pharmaceuticals-16-00299-f011:**
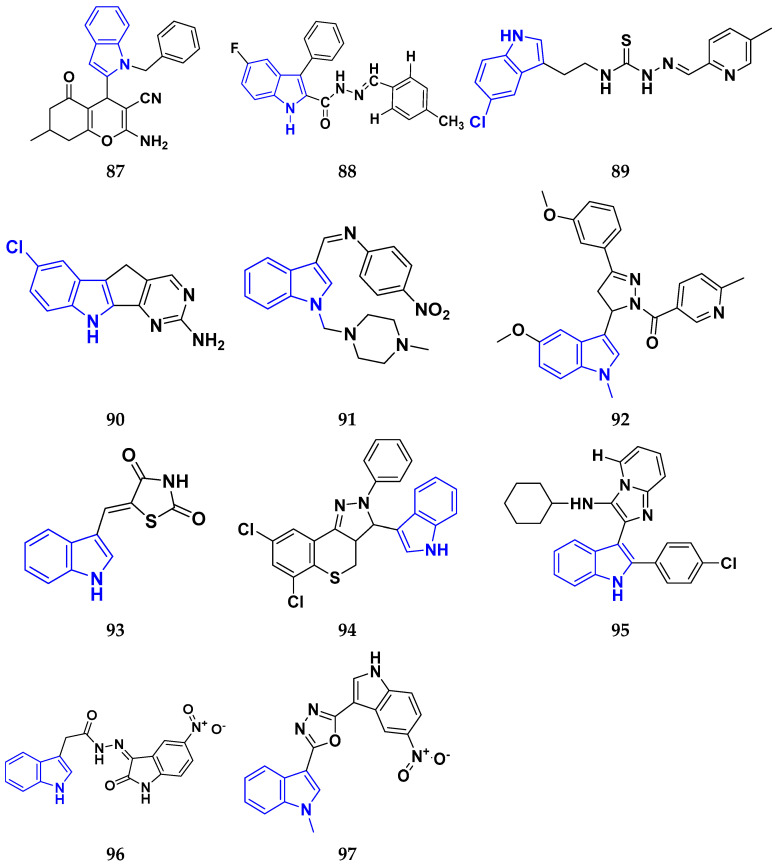
Indole derivatives (**87**–**97**) as anticancer agents.

**Figure 12 pharmaceuticals-16-00299-f012:**
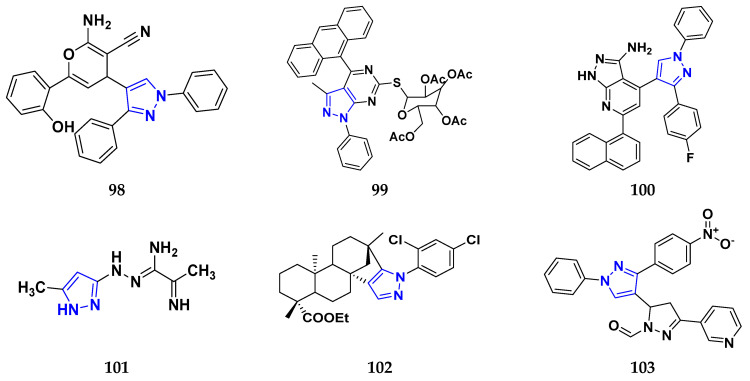
Pyrazole derivatives (**98**–**107**) as anticancer agents.

**Figure 13 pharmaceuticals-16-00299-f013:**
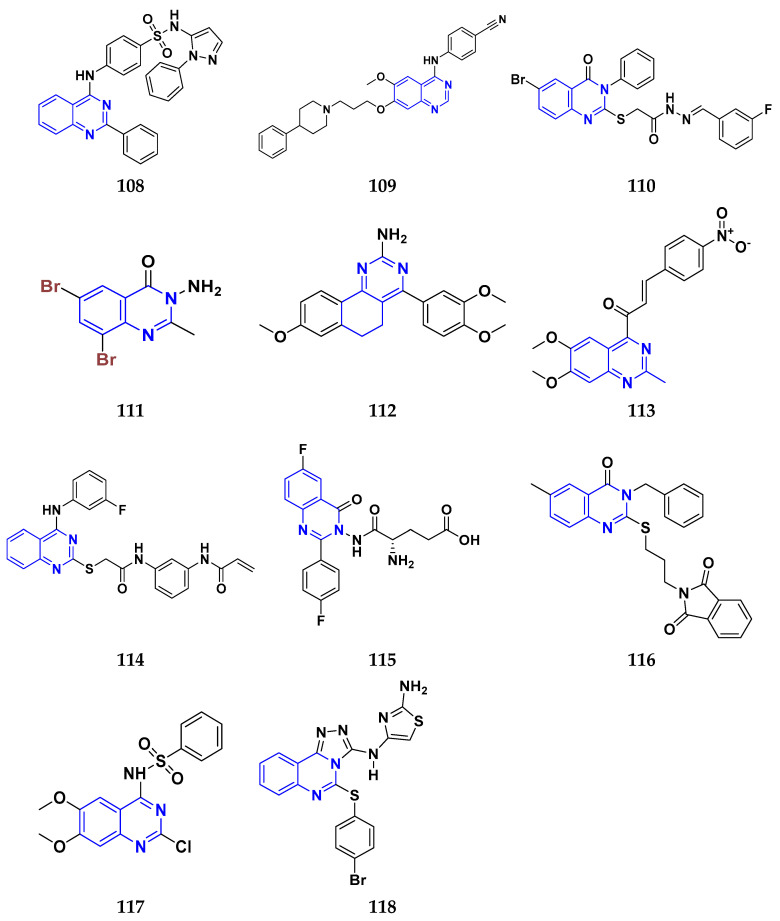
Quinazoline derivatives (**108**–**118**) as anticancer agents.

**Figure 14 pharmaceuticals-16-00299-f014:**
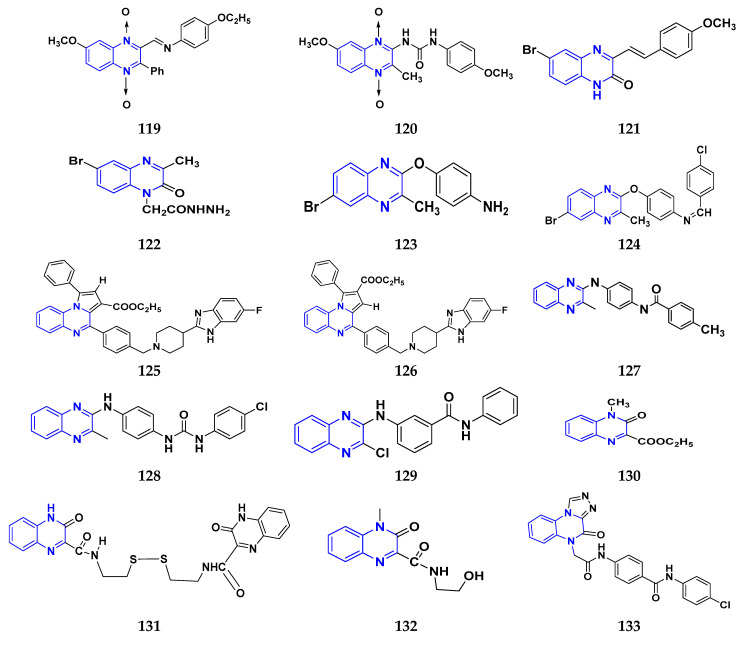
Quinoxaline derivatives (**119**–**143**) as anticancer agents.

**Figure 15 pharmaceuticals-16-00299-f015:**
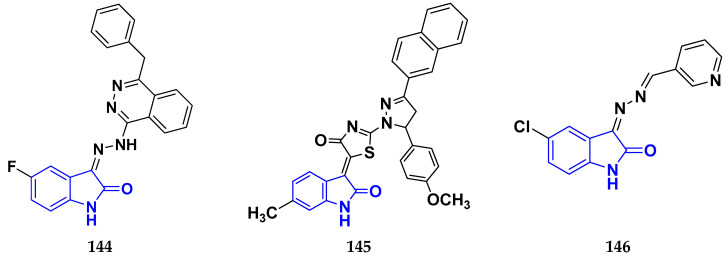
Isatin derivatives (**144**–**150**) as anticancer agents.

**Figure 16 pharmaceuticals-16-00299-f016:**
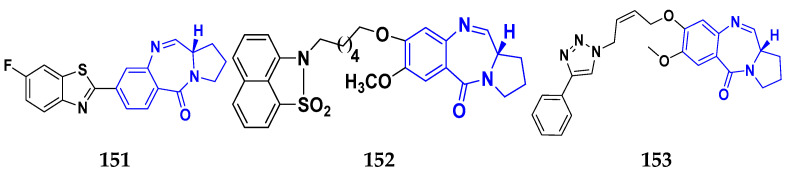
Pyrrolo-benzodiazepines derivatives (**151**–**153**) as anticancer agents.

**Figure 17 pharmaceuticals-16-00299-f017:**
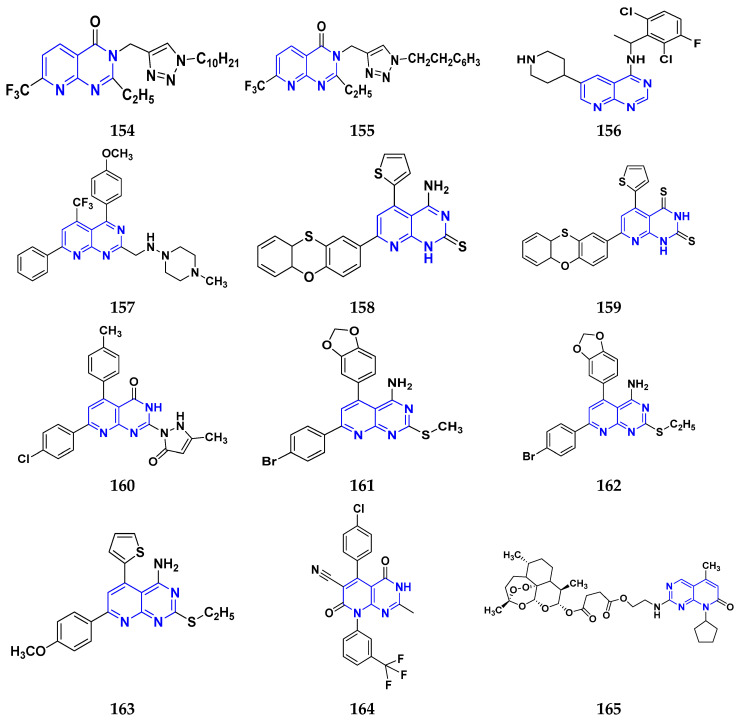
Pyrido[2,3-d] pyrimidine derivatives (**154**–**165**) as anticancer agents.

## Data Availability

All the data given in this manuscript has been taken from published research articles, given in the list of references and are available online.

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
