# Peer review of "Nitrogen Containing Heterocycles as Anticancer Agents: A Medicinal Chemistry Perspective"

_pharmaceuticals, 2023, doi:10.3390/ph16020299_

Round 1

Reviewer 1 Report

Summary of the key contribution of the paper:

The Review Nitrogen containing heterocycles as anticancer agents: A medicinal chemistry perspective is explained the current study deduced that chemically synthesized nitrogen-containing heterocycles (pyrimidine, quinolone, carbazole, pyridine, quinone, imidazole, benzimidazole, hydroxamic acid, triazole, β-lactam, indole, nitrogen mustard, pyrazole, quinazoline, quinoxaline, isatin, pyrrolo benzodiazepines, and pyrido[2,3-d]pyrimidines) are important and versatile molecules against the different forms of cancer. The scope of the review is very interesting, the review can be accepted after addressing the comments below.

Highlights:

·        In this review Several protocols for the synthesis of nitrogen containing heterocycles including pyrimidine, quinolone, carbazole, pyridine, quinone, imidazole, benzimidazole, hydroxamicacid, triazole, β-lactam, indole, nitrogen mustard, pyrazole, quinazoline, quinoxaline, isatin, pyrrolo-benzodiazepines, and pyrido[2,3-d]pyrimidines, which are used in the treatment of different types of cancer, concurrently covering the biochemical mechanisms of action, and cellular targets are reported in the recent few years have been summarized in this review.

·        This review clearly explains that the compiled the FDA approved heterocyclic drugs with nitrogen atom with their pharmacological properties.

·         The figures and tables are well referenced and clear.

·        The resulting review will greatly contribute to academia as well as industry.

·        This study shown the This study will be useful in encouraging the structural based designing of efficacious nitrogen-based drugs for a variety of cancers with minimal side effects.

·        This review will help to inspire more researchers to conceive more fantastic ideas and to find more valuable new synthetic methods.

·        The references are well referenced and clear.

·        Lowlights:

·        The bulk of this review (with exception of anti-cancers applications) focuses on studies done between 2000-2015. It would be helpful to include more recent studies, especially given the use of nitrogen containing heterocycles for treatment of COVID-19.

·        conclusion should be developed more to discuss possible future applications of the suggested modified nitrogen containing heterocycles.

Author Response

Response Letter (Pharmaceuticals-2173501)

Referee 1

In this review Several protocols for the synthesis of nitrogen containing heterocycles including pyrimidine, quinolone, carbazole, pyridine, quinone, imidazole, benzimidazole, hydroxamicacid, triazole, β-lactam, indole, nitrogen mustard, pyrazole, quinazoline, quinoxaline, isatin, pyrrolo-benzodiazepines, and pyrido[2,3-d]pyrimidines, which are used in the treatment of different types of cancer, concurrently covering the biochemical mechanisms of action, and cellular targets are reported in the recent few years have been summarized in this review.

This review clearly explains that the compiled the FDA approved heterocyclic drugs with nitrogen atom with their pharmacological properties.

 The figures and tables are well referenced and clear.

 The resulting review will greatly contribute to academia as well as industry.

This study shown the This study will be useful in encouraging the structural based designing of efficacious nitrogen-based drugs for a variety of cancers with minimal side effects.

This review will help to inspire more researchers to conceive more fantastic ideas and to find more valuable new synthetic methods.

The references are well referenced and clear.

Comment 1.  The bulk of this review (with exception of anti-cancers applications) focuses on studies done between 2000-2015. It would be helpful to include more recent studies, especially given the use of nitrogen containing heterocycles for treatment of COVID-19.

Response to reviewer’s comment: We have compiled nitrogen containing heterocycles as anticancer agents. So, we did not compile the data on use of nitrogen containing heterocycles for treatment of COVID-19.  

Comment 2. Conclusion should be developed more to discuss possible future applications of the suggested modified nitrogen containing heterocycles.

Modification made in revised manuscript: As per the reviewer’s suggestion, we have revised the conclusion in the revised manuscript.

Referee 2

Comment 1. According to the title, the manuscript aims to review nitrogen containing heterocycles as anticancer agents, from a medicinal chemistry perspective. The topic is, thus, very extensive and given that there are already numerous similar review papers, it would be a good idea to immediately clearly indicate what the main focus is and how it is handled. It is not clear who would benefit the most from this review and in what way. It might be useful to create Table of content at the beginning, and a list of abbreviations that would be consistently used in the text. But first, a clear structure of the paper should be made. It should be argued well why certain groups of compounds were selected and clearly explained what the choice of examples used was based on. For example, Section 3.5. has nothing to do with the title of the review – quinones are not nitrogen containing heterocycles and some of the compounds in Figure 7 do not even have a single type of N-containing heterocycle within their structure; same goes for sections 3.8. Hydroxamic acid and 4.2. Nitrogen mustard derivatives. Section 4. is titled “β-lactam derivatives as anticancer agents”, but all other subsections that are part of this section (4.1. Indole, 4.2. Nitrogen mustard, 4.3. Pyrazole, 4.4. Quinazoline etc.) is about groups of compounds that are clearly not β-lactams at all. Throughout the work, listed one after the other, compounds, their IC50 values, cell lines and references and other related information, makes the paper very difficult to read. There is no real discussion or critical review, and the conclusion is too general. For example, from what can it be concluded that such compounds would allow fewer side effects in anticancer therapies? How additional groups affect activities of the reviewed compounds?

Modification made in revised manuscript: As per the reviewer’s suggestion, we have added the table of contents and abbreviation’s. Also, we have removed the hydroxamic acid, Quinone and nitrogen mustard and its relevant data in the revised manuscript. Also we have revised the numbering of headings and conclusion in the revised manuscript. As this review has extensively covered N-containing heterocyclic anticancer compounds, the academic as well as industrial researchers working in the design and development of novel anticancer agents will be benefited.  This review has compiled N-containing heterocyclic anticancer compounds and only most potent compounds from individual series have been reported as examples. At the end of every scaffold, we have highlighted the promising structural features responsible for anticancer activity of that scaffold. Conclusion of the manuscript is also revised.

Comment 2. Line 104 - check commas in number (2,331,700).

Modification made in revised manuscript: As per the reviewer’s suggestion, we have revised the commas in number in revised manuscript.

Comment 3. Line 105 and Fig.1 correct lop P to log P.

Modification made in revised manuscript: As per the reviewer’s suggestion, we have revised the correct lop P to log P in revised manuscript.

Comment 4. Comment on violating RO5 significance in Fig.2.

Modification made in revised manuscript: As per the reviewer’s suggestion, we have added the comment on violating RO5 significance in revised manuscript.

Comment 5. Line 120-122 and Table 1 - indicate what was the criterion for the selection of listed molecules.

Response to reviewer’s comment: we have randomly selected the nitrogen containing FDA approved drugs as anticancer agents.

Comment 6. Table 1 – under Drug target, why some entries are underlined (e.g. entry 40) or italicized (e.g. entry 47)?

Modification made in revised manuscript: As per the reviewer’s suggestion, we have revised the text in table 1.

Comment 7. Table 1 - Oxaliplatin (entry 25) is not a heterocycle with nitrogen atom(s) within a ring.

Modification made in revised manuscript: As per the reviewer’s suggestion, we have revised the structure of Oxaliplatin in revised manuscript.

Comment 8. Line 124 - What is meant by “Current advances…”? Please clarify.

Response to reviewer’s comment: “Current advances”means we have compiled the latest articles related to our topic of the study.

Comment 9. Line 158 – “Two…” (Capital letter).

Modification made in revised manuscript: As per the reviewer’s suggestion, we have revised the text in revised manuscript.

Comment 10. Line 186 – compounds (remove capital letter).

Modification made in revised manuscript: As per the reviewer’s suggestion, we have revised the text in revised manuscript.

Comment 11. Line 292 – colchicine (remove capital letter - check also throughout the text unjustified use of capital letter, in many places e.g. Synthesis).

Modification made in revised manuscript: As per the reviewer’s suggestion, we have revised the text in revised manuscript.

Comment 12. Line 447 – ‘were tested’ instead ‘was tested’.

Modification made in revised manuscript: As per the reviewer’s suggestion, we have revised the text in revised manuscript.

Comment 13. Check the structures in all figures and tables because some are stretched.

Modification made in revised manuscript: As per the reviewer’s suggestion, we have checked the structures in all figures and tables text of revised manuscript.

Comment 14. It would be useful to use bold numbers for compounds in text and figures to make easier finding the relevant information.

Modification made in revised manuscript: As per the reviewer’s suggestion, we have changed the compound no in text and figure of revised manuscript.

Authors are highly thankful to all the reviewer’s for their valuable suggestions to improve the quality of manuscript.

Reviewer 2 Report

According to the title, the manuscript aims to review nitrogen containing heterocycles as anticancer agents, from a medicinal chemistry perspective. The topic is, thus, very extensive and given that there are already numerous similar review papers, it would be a good idea to immediately clearly indicate what the main focus is and how it is handled. It is not clear who would benefit the most from this review and in what way. It might be useful to create Table of content at the beginning, and a list of abbreviations that would be consistently used in the text. But first, a clear structure of the paper should be made. It should be argued well why certain groups of compounds were selected and clearly explained what the choice of examples used was based on. For example, Section 3.5. has nothing to do with the title of the review – quinones are not nitrogen containing heterocycles and some of the compounds in Figure 7 do not even have a single type of N-containing heterocycle within their structure; same goes for sections 3.8. Hydroxamic acid and 4.2. Nitrogen mustard derivatives. Section 4. is titled “β-lactam derivatives as anticancer agents”, but all other subsections that are part of this section (4.1. Indole, 4.2. Nitrogen mustard, 4.3. Pyrazole, 4.4. Quinazoline etc.) is about groups of compounds that are clearly not β-lactams at all.

Throughout the work, listed one after the other, compounds, their IC50 values, cell lines and references and other related information, makes the paper very difficult to read. There is no real discussion or critical review, and the conclusion is too general. For example, from what can it be concluded that such compounds would allow fewer side effects in anticancer therapies? How additional groups affect activities of the reviewed compounds?

Line 104 - check commas in number (2,331,700).

Line 105 and Fig.1 correct lop P to log P.

Comment on violating RO5 significance in Fig.2.

Line 120-122 and Table 1 - indicate what was the criterion for the selection of listed molecules.

Table 1 – under Drug target, why some entries are underlined (e.g. entry 40) or italicized (e.g. entry 47)?

Table 1 - Oxaliplatin (entry 25) is not a heterocycle with nitrogen atom(s) within a ring.

Line 124 - what is meant by “Current advances…”? Please clarify.

Line 158 – “Two…” (capital letter).

Line 186 – compounds (remove capital letter).

Line 292 – colchicine (remove capital letter - check also throughout the text unjustified use of capital letter, in many places e.g. Synthesis).

Line 447 – ‘were tested’ instead ‘was tested’.

Check the structures in all figures and tables because some are stretched.

It would be useful to use bold numbers for compounds in text and figures to make easier finding the relevant information.

Author Response

(The authors gave the same response as above.)

Round 2

Reviewer 2 Report

The manuscript is improved, but it is necessary to check the structures and for spelling errors once again. I would still recommend that a critical review be stepped up before publication and more specific conclusions should be drawn about individual groups of compounds and their biological (anticancer) activity.

Author Response

Response Letter (pharmaceuticals-2173501)

Referee 1

The manuscript is improved, but it is necessary to check the structures and for spelling errors once again. I would still recommend that a critical review be stepped up before publication and more specific conclusions should be drawn about individual groups of compounds and their biological (anticancer) activity.

Modification made in revised manuscript: As per the reviewer’s suggestion, we have checked all the structures and revised the manuscript for grammatical and typographical errors. We have added the structural features of potent compounds in discussion at the end of every scaffold. Also we have added the discussion of potent compounds from each scaffold and their specific anticancer activity in the conclusion.

Once again, authors are highly thankful to all the reviewer’s for their valuable suggestions to improve the quality of manuscript.